# A Theory of Usable Information under Computational Constraints

**Yilun Xu**
CFCS, Peking University
xuyilun@pku.edu.cn

**Shengjia Zhao**
Stanford University
sjzhao@stanford.edu

**Jiaming Song**
Stanford University
tsong@cs.stanford.edu

**Russell Stewart**
russell.sb.nebel@gmail.com

**Stefano Ermon**
Stanford University
ermon@cs.stanford.edu

## Abstract

We propose a new framework for reasoning about information in complex systems. Our foundation is based on a variational extension of Shannon's information theory that takes into account the modeling power and computational constraints of the observer. The resulting *predictive $\mathcal{V}$-information* encompasses mutual information and other notions of informativeness such as the coefficient of determination. Unlike Shannon's mutual information and in violation of the data processing inequality, $\mathcal{V}$-information can be created through computation. This is consistent with deep neural networks extracting hierarchies of progressively more informative features in representation learning. Additionally, we show that by incorporating computational constraints, $\mathcal{V}$-information can be reliably estimated from data even in high dimensions with PAC-style guarantees. Empirically, we demonstrate predictive $\mathcal{V}$-information is more effective than mutual information for structure learning and fair representation learning.

## 1 Introduction

Extracting actionable *information* from noisy, possibly redundant, and high-dimensional data sources is a key computational and statistical challenge at the core of AI and machine learning. Information theory, which lies at the foundation of AI and machine learning, provides a conceptual framework to characterize information in a mathematically rigorous sense (Shannon & Weaver, 1948; Cover & Thomas, 1991). However, important computational aspects are not considered in information theory. To illustrate this, consider a dataset of encrypted messages intercepted from an opponent. According to information theory, these encrypted messages have high mutual information with the opponent's plans. Indeed, with infinite computation, the messages can be decrypted and the plans revealed. Modern cryptography originated from this observation by Shannon that perfect secrecy is (essentially) impossible if the adversary is computationally unbounded (Shannon & Weaver, 1948). This motivated cryptographers to consider restricted classes of adversaries that have access to limited computational resources (Pass & Shelat, 2010). More generally, it is known that information theoretic quantities can be expressed in terms of betting games (Cover & Thomas, 1991). For example, the (conditional) entropy of a random variable $X$ is directly related to how predictable $X$ is in a certain betting game, where an agent is rewarded for correct guesses. Yet, the standard definition unrealistically assumes agents are computationally unbounded, i.e., they can employ arbitrarily complex prediction schemes.

Leveraging modern ideas from variational inference and learning (Ranganath et al., 2013; Kingma & Welling, 2013; LeCun et al., 2015), we propose an alternative formulation based on realistic computational constraints that is in many ways closer to our intuitive notion of information, which we term *predictive $\mathcal{V}$-information*. Without constraints, predictive $\mathcal{V}$-information specializes to classic mutual information. Under natural restrictions, $\mathcal{V}$-information specializes to other well-known notions of predictiveness, such as the coefficient of determination ($R^2$). A consequence of this new formulation is that computation can "create usable information" (e.g., by decrypting the intercepted messages), invalidating the famous data processing inequality. This generalizes the idea that clever

feature extraction enables prediction with extremely simple (e.g., linear) classifiers, a key notion in modern representation and deep learning (LeCun et al., 2015).

As an additional benefit, we show that predictive $\mathcal{V}$-information can be estimated with statistical guarantees using the Probably Approximately Correct framework (Valiant, 1984). This is in sharp contrast with Shannon information, which is well known to be difficult to estimate for high dimensional or continuous random variables (Battiti, 1994). Theoretically we show that the statistical guarantees of estimating $\mathcal{V}$ information translate to statistical guarantees for a variant of the Chow-Liu algorithm for structure learning. In practice, when the observer employs deep neural networks as a prediction scheme, $\mathcal{V}$-information outperforms methods that approximate Shannon information in various applications, including Chow-Liu tree contruction in high dimension and gene regulatory network inference.

## 2 DEFINITIONS AND NOTATIONS

To formally define the predictive $\mathcal{V}$-information, we begin with a formal model of a computationally bounded agent trying to predict the outcome of a real-valued random variable $Y$; the agent is either provided another real-valued random variable $X$ as side information, or provided no side information $\varnothing$. We use $\mathcal{X}$ and $\mathcal{Y}$ to denote the samples spaces of $X$ and $Y$ respectively (while assuming they are separable), and use $\mathcal{P}(\mathcal{X})$ to denote the set of all probability measures over the Borel algebra on $\mathcal{X}$ ($\mathcal{P}(\mathcal{Y})$ similarly defined for $\mathcal{Y}$).

**Definition 1** (Predictive Family). [1] *Let $\Omega = \{f : \mathcal{X} \cup \{\varnothing\} \to \mathcal{P}(\mathcal{Y})\}$. We say that $\mathcal{V} \subseteq \Omega$ is a predictive family if it satisfies*

$$\forall f \in \mathcal{V}, \forall P \in \text{range}(f), \quad \exists f' \in \mathcal{V}, \quad s.t. \quad \forall x \in \mathcal{X}, f'[x] = P, f'[\varnothing] = P \tag{1}$$

A predictive family is a set of predictive models the agent is allowed to use, e.g., due to computational or statistical constraints. We refer to the additional condition in Eq.(1) as *optional ignorance*. Intuitively, it means that the agent can, in the context of the prediction game we define next, ignore the side information if she chooses to.

**Definition 2** (Predictive conditional $\mathcal{V}$-entropy). *Let $X, Y$ be two random variables taking values in $\mathcal{X} \times \mathcal{Y}$, and $\mathcal{V}$ be a predictive family. Then the predictive conditional $\mathcal{V}$-entropy is defined as*

$$H_{\mathcal{V}}(Y|X) = \inf_{f \in \mathcal{V}} \mathbb{E}_{x,y \sim X,Y} \left[ -\log f[x](y) \right]$$

$$H_{\mathcal{V}}(Y|\varnothing) = \inf_{f \in \mathcal{V}} \mathbb{E}_{y \sim Y} \left[ -\log f[\varnothing](y) \right]$$

*We additionally call $H_{\mathcal{V}}(Y|\varnothing)$ the $\mathcal{V}$-entropy, and also denote it as $H_{\mathcal{V}}(Y)$*

In our notation $f$ is a function $\mathcal{X} \cup \{\varnothing\} \to \mathcal{P}(\mathcal{Y})$, so $f[x] \in \mathcal{P}(\mathcal{Y})$ is a probability measure on $\mathcal{Y}$ chosen based on the received side information $x$ (we use $f[\cdot]$ instead of the more conventional $f(\cdot)$); and $f[x](y) \in \mathbb{R}$ is the value of the density evaluated at $y \in \mathcal{Y}$. Intuitively, $\mathcal{V}$ (conditional) entropy is the smallest expected negative log-likelihood that can be achieved predicting $Y$ given observation (side information) $X$ (or no side information $\varnothing$), using models from $\mathcal{V}$. Eq.(1) means that whenever the agent can use $P$ to predict $\mathcal{Y}$'s outcomes, it has the option to ignore the input, and use $P$ no matter whether $X$ is observed or not.

Definition 2 generalizes several known definitions of uncertainty. For example, as shown in proposition 2, if the $\mathcal{V}$ is the largest possible predictive family that includes all possible models, i.e. $\mathcal{V} = \Omega$, then Definition 2 reduces to Shannon entropy: $H_{\Omega}(Y|X) = H(Y|X)$ and $H_{\mathcal{V}}(Y|\varnothing) = H_{\Omega}(Y) = H(Y)$. By choosing more restrictive families $\mathcal{V}$, we recover several other notions of uncertainty such as trace of covariance, as will be shown in Proposition 1.

Shannon mutual information is a measure of changes in entropy when conditioning on new variables:

$$I(X;Y) = H(Y) - H(Y|X) = H_{\Omega}(Y) - H_{\Omega}(Y|X) \tag{2}$$

Here, we will use predictive $\mathcal{V}$-entropy to define an analogous quantity, $I_{\mathcal{V}}(X \to Y)$, to represent the *change* in predictability of an output variable $Y$ when given side information $X$.

---

[1]Regularity Conditions: To minimize technical overhead we restrict out discussion only to distributions with probability density functions (PDF) or probability mass functions (PMF) with respect to the underlying measure. Also $\varnothing \notin \mathcal{X}$.

**Definition 3** (Predictive $\mathcal{V}$-information)**.** *Let $X, Y$ be two random variables taking values in $\mathcal{X} \times \mathcal{Y}$, and $\mathcal{V}$ be a predictive family. The predictive $\mathcal{V}$-information from $X$ to $Y$ is defined as*

$$I_\mathcal{V}(X \to Y) = H_\mathcal{V}(Y|\varnothing) - H_\mathcal{V}(Y|X) \tag{3}$$

## 2.1 IMPORTANT SPECIAL CASES

Several important notions of uncertainty and predictiveness are special cases of our definition. Note that when we are defining $\mathcal{V}$-entropy of a random variable $Y$ in sample space $\mathcal{Y} \in \mathbb{R}^d$ (without side information), out of convenience we can assume $\mathcal{X}$ is empty $\mathcal{X} = \varnothing$ (this does not violate our requirement that $\varnothing \notin \mathcal{X}$.)

**Proposition 1.** *For $\mathcal{V}$-entropy and $\mathcal{V}$-information, we have*

1. *Let $\Omega$ be as in Def. 1. Then $H_\Omega(Y)$ is the Shannon entropy, $H_\Omega(Y \mid X)$ is the Shannon conditional entropy, and $I_\Omega(Y \to X)$ is the Shannon mutual information.*

2. *Let $\mathcal{Y} = \mathbb{R}^d$ and $\mathcal{V} = \{f : \{\varnothing\} \to P_\mu \mid \mu \in \mathbb{R}^d\}$, where $P_\mu$ is the distribution with density $y \mapsto \frac{1}{Z} e^{-\|y-\mu\|_2}$ where $Z = \int e^{-\|y-\mu\|_2} dy$, then the $\mathcal{V}$-entropy of a random variable $Y$ equals its mean absolute deviation, up to an additive constant.*

3. *Let $\mathcal{Y} = \mathbb{R}^d$ and $\mathcal{V} = \{f : \{\varnothing\} \to \mathcal{N}(\mu, \Sigma) \mid \mu \in \mathbb{R}^d, \Sigma = 1/2 I_{d \times d}\}$, then the $\mathcal{V}$-entropy of a random variable $Y$ equals the trace of its covariance $\operatorname{tr}(\operatorname{Cov}(Y))$, up to an additive constant.*

4. *Let $\mathcal{V} = \{f : \{\varnothing\} \to Q_{\mathbf{t},\theta}, \theta \in \Theta\}$, where $Q_{\mathbf{t},\theta}$ is a distribution in a minimal exponential family with sufficient statistics $\mathbf{t} : \mathcal{Y} \to \mathbb{R}^d$ and set of natural parameters $\Theta$. For a random variable $Y$ with expected sufficient statistics $\mu_Y = \mathbb{E}[\mathbf{t}(Y)]$, the $\mathcal{V}$-entropy of $Y$ is the maximum Shannon entropy over all random variables $\hat{Y}$ with identical expected sufficient statistics, i.e. $\mathbb{E}[\mathbf{t}(\hat{Y})] = \mu_Y$.*

5. *Let $\mathcal{Y} = \mathbb{R}^d$, $\mathcal{X}$ be any vector space, and $\mathcal{V} = \{f : x \mapsto \mathcal{N}(\phi(x), \Sigma), x \in \mathcal{X}; \varnothing \mapsto \mathcal{N}(\mu, \Sigma) | \mu \in \mathbb{R}^d; \Sigma = 1/2 I_{d \times d}, \phi \in \Phi\}$, where $\Phi$ is the set of linear functions $\{\phi : \mathcal{X} \to \mathbb{R}^d\}$, then $\mathcal{V}$-information $I_\mathcal{V}(X \to Y)$ equals the (unnormalized) maximum coefficient of determination $R^2 \cdot \operatorname{tr}(\operatorname{Cov}(Y))$ for linear regression.*

The trace of covariance represents a natural notion of uncertainty – for example, a random variable with zero variance (when $d = 1, \operatorname{tr}(\operatorname{Cov}(Y)) = \operatorname{Var}(Y))$) is trivial to predict. Proposition 1.3 shows that the trace of covariance corresponds to a notion of surprise (in the Shannon sense) for an agent *restricted* to make predictions using certain Gaussian models. More broadly, a similar analogy can be drawn for other exponential families of distributions. In the same spirit, the coefficient of determination, also known as the fraction of variance explained, represents a natural notion of informativeness for computationally bounded agents. Also note that in the case of Proposition 1.4, the $\mathcal{V}$-entropy is invariant if the expected sufficient statistics remain the same.

# 3 PROPERTIES OF $\mathcal{V}$-INFORMATION

## 3.1 ELEMENTARY PROPERTIES

We first show several elementary properties of $\mathcal{V}$-entropy and $\mathcal{V}$-information. In particular, $\mathcal{V}$-information preserves many properties of Shannon information that are desirable in a machine learning context. For example, mutual information (and $\mathcal{V}$-information) should be non-negative as conditioning on additional side information $X$ should not reduce an agent's ability to predict $Y$.

**Proposition 2.** *Let $Y$ and $X$ be any random variables on $\mathcal{Y}$ and $\mathcal{X}$, and $\mathcal{V}$ and $\mathcal{U}$ be any predictive families, then we have*

1. **Monotonicity***: If $\mathcal{V} \subseteq \mathcal{U}$, then $H_\mathcal{V}(Y) \geq H_\mathcal{U}(Y)$, $H_\mathcal{V}(Y \mid X) \geq H_\mathcal{U}(Y \mid X)$.*

2. **Non-Negativity***: $I_\mathcal{V}(X \to Y) \geq 0$.*

3. **Independence***: If $X$ is independent of $Y$, $I_\mathcal{V}(X \to Y) = I_\mathcal{V}(Y \to X) = 0$.*

The *optional ignorance* requirement in Eq.(1) is a technical condition needed for these properties to hold. Intuitively, it guarantees that conditioning on side information does not restrict the class of densities the agent can use to predict $Y$. This property is satisfied by many existing machine learning models, often by setting some weights to zero so that an input is effectively ignored.

### 3.2 ON THE PRODUCTION OF INFORMATION THROUGH PREPROCESSING

The Data Processing Inequality guarantees that computing on data cannot increase its mutual information with other random variables. Formally, letting $t : \mathcal{X} \to \mathcal{X}$ be any function, $t(X)$ cannot have higher mutual information with $Y$ than $X$: $I(t(X); Y) \leq I(X; Y)$. But is this property desirable? In analyzing optimal communication, yes - it demonstrates a fundamental limit to the number of bits that can be transmitted through a communication channel. However, we argue that in machine learning settings this property is less appropriate.

Consider an RSA encryption scheme where the public key is known. Given plain text and its corresponding encrypted text $X$, if we have infinite computation, we can perfectly compute one from the other. Therefore, the plain text and the encrypted text should have identical Shannon mutual information with respect to any label $Y$ we want to predict. However, to any human (or machine learning algorithm), it is certainly easier to predict the label from the plain text than the encrypted text. In other words, decryption increases a human's ability to predict the label: processing increases the "usable information". More formally, denoting $t$ as the decryption algorithm and $\mathcal{V}$ as a class of natural language processing functions, we have that: $I_{\mathcal{V}}(t(X) \to Y) > I_{\mathcal{V}}(X \to Y) \approx 0$.

As another example, consider the mutual information between an image's pixels and its label. Due to data processing inequality, we cannot expect to use a function to map raw pixels to "features" that have higher mutual information with the label. However, the fundamental principle of representation learning is precisely the ability to learn predictive features — functions of the raw inputs that enable predictions with higher accuracy. Because of this key difference between $\mathcal{V}$-information and Shannon information, machine learning practices such as representation learning can be justified in the information theoretic context.

### 3.3 ON THE ASYMMETRY OF PREDICTIVE $\mathcal{V}$-INFORMATION

$\mathcal{V}$-information also captures the intuition that sometimes, it is easy to predict $Y$ from $X$ but not vice versa. In fact, modern cryptography is founded on the assumption that certain functions $h : \mathcal{X} \to \mathcal{Y}$ are one-way, meaning that there exists an polynomial algorithm to compute $h(x)$ but no polynomial algorithm to compute $h^{-1}(y)$. This means that if $\mathcal{V}$ contains all polynomial-time computable functions, then $I_{\mathcal{V}}(X \to h(X)) \gg I_{\mathcal{V}}(h(X) \to X)$.

This property is also reasonable in the machine learning context. For example, several important methods for causal discovery (Peters et al., 2017) rely on this asymmetry: if $X$ causes $Y$, then usually it is easier to predict $Y$ from $X$ than vice versa; another commonly used assumption is that $Y|X$ can be accurately modeled by a Gaussian distribution, while $X|Y$ cannot (Pearl, 2000).

## 4 PAC GUARANTEES FOR $\mathcal{V}$-INFORMATION ESTIMATION

For many practical applications of mutual information (e.g., structure learning), we do not know the joint distribution of $X, Y$, so cannot directly compute the mutual information. Instead we only have samples $\{(x_i, y_i)\}_{i=1}^N \sim X, Y$ and need to estimate mutual information from data.

Shannon information is notoriously difficult to estimate for high dimensional random variables. Although non-parametric estimators of mutual information exist (Kraskov et al., 2004; Darbellay & Vajda, 1999; Gao et al., 2017), these estimators do not scale to high dimensions. Several variational estimators for Shannon information have been recently proposed (van den Oord et al., 2018; Nguyen et al., 2010; Belghazi et al., 2018), but have two shortcomings: due to their variational assumptions, their bias/variance tradeoffs are poorly understood and they are still not efficient enough for high dimensional problems. For example, the CPC estimator suffers from large bias, since its estimates saturate at $\log N$ where $N$ is the batch size (van den Oord et al., 2018; Poole et al., 2019); the NWJ estimator suffers from large variance that grows at least exponentially in the ground-truth mutual information (Song & Ermon, 2019). Please see Appendix B for more details and proofs.

On the other hand, $\mathcal{V}$-information is explicit about the assumptions (as a feature instead of a bug). $\mathcal{V}$-information is also easy to estimate with guarantees if we can bound the complexity of $\mathcal{V}$ (such as its Radamacher or covering number complexity) As we will show, bounds on the complexity of $\mathcal{V}$ directly translate to PAC (Valiant, 1984) bounds for $\mathcal{V}$-information estimation. In practice, we can efficiently optimize over $\mathcal{V}$, e.g., via gradient descent. In this paper we will present the Rademacher complexity version; other complexity measures (such as covering number) can be derived similarly.

**Definition 4** (Empirical $\mathcal{V}$-information). *Let $X, Y$ be two random variables taking values in $\mathcal{X}, \mathcal{Y}$ and $\mathcal{D} = \{(x_i, y_i)\}_{i=1}^{N} \sim X, Y$ denotes the set of samples drawn from the joint distribution over $\mathcal{X}$ and $\mathcal{Y}$. $\mathcal{V}$ is a predictive family. The empirical $\mathcal{V}$-information (under $\mathcal{D}$) is the following $\mathcal{V}$-information under the empirical distribution defined via $\mathcal{D}$:*

$$\hat{I}_{\mathcal{V}}(X \to Y; \mathcal{D}) = \inf_{f \in \mathcal{V}} \frac{1}{|\mathcal{D}|} \sum_{y_i \in \mathcal{D}} \log \frac{1}{f[\varnothing](y_i)} - \inf_{f \in \mathcal{V}} \frac{1}{|\mathcal{D}|} \sum_{x_i, y_i \in \mathcal{D}} \log \frac{1}{f[x_i](y_i)} \tag{4}$$

Then we have the following PAC bound over the empirical $\mathcal{V}$-information:

**Theorem 1.** *Assume $\forall f \in \mathcal{V}, x \in \mathcal{X}, y \in \mathcal{Y}, \log f[x](y) \in [-B, B]$. Then for any $\delta \in (0, 0.5)$, with probability at least $1 - 2\delta$, we have:*

$$\left| I_{\mathcal{V}}(X \to Y) - \hat{I}_{\mathcal{V}}(X \to Y; \mathcal{D}) \right| \leq 4 \mathfrak{R}_{|\mathcal{D}|}(\mathcal{G}_{\mathcal{V}}) + 2B \sqrt{\frac{2 \log \frac{1}{\delta}}{|\mathcal{D}|}} \tag{5}$$

*where we define the function family $\mathcal{G}_{\mathcal{V}} = \{g | g(x, y) = \log f[x](y), f \in \mathcal{V}\}$, and $\mathfrak{R}_N(\mathcal{G})$ denotes the Rademacher complexity of $\mathcal{G}$ with sample number $N$.*

Typically, the Rademacher complexity term satisfies $\mathfrak{R}_{|\mathcal{D}|}(\mathcal{G}_{\mathcal{V}}) = \mathcal{O}(|\mathcal{D}|^{-\frac{1}{2}})$ (Bartlett & Mendelson, 2001; Gao & Zhou, 2016). It's worth noticing that a complex function family $\mathcal{V}$ (i.e., with large Rademacher complexity) could lead to overfitting. On the other hand, an overly-simple $\mathcal{V}$ may not be expressive enough to capture the relationship between $X$ and $Y$. As an example of the theorem, we provide a concrete estimation bound when $\mathcal{V}$ is chosen to be linear functions mapping $\mathcal{X}$ to the mean of a Gaussian distribution. This was shown in Proposition 1 to lead to the coefficient of determination.

**Corollary 1.1.** *Assume $\mathcal{X} = \{x \in \mathbb{R}^{d_x}, \|x\|_2 \leq k_x\}$ and $\mathcal{Y} = \{y \in \mathbb{R}^{d_y}, \|y\|_2 \leq k_y\}$. If*

$$\mathcal{V} = \{f : f[x] = \mathcal{N}(Wx + b, I), f[\varnothing] = \mathcal{N}(c, I), W \in \mathbb{R}^{d_y \times d_x}, b, c \in \mathbb{R}^{d_y}, \|(W, b)\|_2 \leq 1\}$$

*Denote $M = (k_x + k_y)^2 + \log 2\pi$, then $\forall \delta \in (0, 0.5)$, with probability at least $1 - 2\delta$:*

$$\left| I_{\mathcal{V}}(X \to Y) - \hat{I}_{\mathcal{V}}(X \to Y; \mathcal{D}) \right| \leq \frac{M}{\sqrt{4|\mathcal{D}|}} \left( 1 + 4 \sqrt{2 \log \frac{1}{\delta}} \right)$$

Similar results can be obtained using other classes of machine learning models with known (Rademacher) complexity.

## 5 STRUCTURE LEARNING WITH $\mathcal{V}$-INFORMATION

Among many possible applications of $\mathcal{V}$-information, we show how to use it to perform structure learning with provable guarantees. The goal of structure learning is to learn a directed graphical model (Bayesian network) or undirected graphical model (Markov network) that best captures the (conditional) independence structure of an underlying data generating process. Structure learning is difficult in general, but if we restrict ourselves to certain set of graphs $G$, there are efficient algorithms. In particular, the Chow-Liu algorithm (Chow & Liu, 1968) can efficiently learn tree graphs (i.e. $G$ is the set of trees). Chow & Liu (1968) show that the problem can be reduced to:

$$g^* = \arg\max_{g \in G_{\text{tree}}} \sum_{(X_i, X_j) \in \text{edge}(g)} I(X_i, X_j) \tag{6}$$

where $I(X_i, X_j)$ is the Shannon mutual information between variables $X_i$ and $X_j$. In other words, it suffices to construct the maximal weighted spanning tree where the weight between two vertices is

their Shannon mutual information. Chow & Wagner (1973) show that the Chow-Liu algorithm is consistent, i.e, it recovers the true solution as the dataset size goes to infinity. However, the finite sample behavior of the Chow-Liu algorithm for high dimensional problems is much less studied, due to the difficulty of estimating mutual information. In fact, we show in our experiments that the empirical performance is often poor, even with state-of-the-art estimators. Additionally, methods based on mutual information cannot take advantage of intrinsically asymmetric relationships, which are common for example in gene regulatory networks (Meyer et al., 2007).

To address these issues, we propose a new structure learning algorithm based on $\mathcal{V}$-information instead of Shannon information. The idea is that we can associate to each *directed* edge in $G$ (i.e., each pair of variables) a suitable predictive family $\mathcal{V}_{i,j}$ (cf. Def 1). The main challenge is that we cannot simply replace mutual information with $\mathcal{V}$-information in Eq. 6 because $\mathcal{V}$-information is asymmetric – we now have to optimize over *directed* trees:

$$g^* = \arg\max_{g \in G_{\mathrm{d-tree}}} \sum_{i=2}^{m} I_{\mathcal{V}_{t(g)(i),i}}(X_{t(g)(i)} \to X_i) \tag{7}$$

where $G_{\mathrm{d-tree}}$ is the set of directed trees, and $t(g) : \mathbb{N} \to \mathbb{N}$ is the function mapping each non-root node of directed tree $g$ to its parent, and $\mathcal{V}_{i,j}$ is the predictive family for random variables $X_i$ and $X_j$. After estimating $\mathcal{V}$-information on each edge, we use the Chu-Liu algorithm (Chu & Liu, 1965) to construct the *maximal directed spanning tree*. This allows us to solve (7) exactly, even though there is a combinatorially large number of trees to consider. Pseudocode is summarized in Algorithm 1 in Appendix. Denote $C(g) = \sum_{i=2}^{m} I_{\mathcal{V}_{t(g)(i),i}}(X_{t(g)(i)} \to X_i)$, we show in the following theorem that unlike the original Chow-Liu algorithm, our algorithm has guarantees in the finite samples regime, even in continuous settings:

**Theorem 2.** *Let $\{X_i\}_{i=1}^{m}$ be the set of m random variables, $\mathcal{D}_{i,j}$ (resp. $\mathcal{D}_j$) be the set of samples drawn from $P(X_i, X_j)$ (resp. $P(X_j)$). Denote the optimal directed tree with maximum expected edge weights sum $C(g)$ as $g^*$ and the optimal directed tree constructed on the dataset $\mathcal{D}$ as $\hat{g}$. Then with the assumption in theorem 1, for any $\delta \in (0, \frac{1}{2m(m-1)})$, with probability at least $1 - 2m(m-1)\delta$, we have:*

$$C(\hat{g}) \geq C(g^*) - 2(m-1) \max_{i,j} \left\{ 2\Re_{\mathcal{D}_{i,j}}(\mathcal{G}_{\mathcal{V}_{i,j}}) + 2\Re_{\mathcal{D}_j}(\mathcal{G}_{\mathcal{V}_j}) + B\sqrt{2\log\frac{1}{\delta}}(|\mathcal{D}_j|^{-\frac{1}{2}} + |\mathcal{D}_{i,j}|^{-\frac{1}{2}}) \right\} \tag{8}$$

Theorem 2 shows that the total edge weights of the maximal directed spanning tree constructed by algorithm 1 would be close to the optimal total edge weights if the Rademacher term is small. Although larger $C(g)$ does not necessarily lead to better Chow-Liu trees, empirically we find that the optimal tree in the sense of equation (7) is consistent with the optimal tree in equation (6) under commonly used $\mathcal{V}$.

## 6 EXPERIMENTAL RESULTS

### 6.1 STRUCTURE LEARNING WITH CONTINUOUS HIGH-DIMENSIONAL DATA

We generate synthetic data using various ground-truth tree structures $g^*$ with between 7 and 20 variables, where each variable is 10-dimensional. We use Gaussians, Exponentials, and Uniforms as ground truth edge-conditionals. We use $\mathcal{V}$-information(Gaussian) and $\mathcal{V}$-information(Logistic) to denote Algorithm 1 with two different $\mathcal{V}$ families. Please refer to Appendix D.1 for more details. We compare with the original Chow-Liu algorithm equipped with state-of-the-art mutual information estimators: **CPC** (van den Oord et al., 2018), **NWJ** (Nguyen et al., 2010) and **MINE** (Belghazi et al., 2018), with the same neural network architecture as the $\mathcal{V}$-families for fair comparison. All the experiments are repeated for 10 times. As a performance metric, we use the wrong-edges-ratio (the ratio of edges that are different from ground truth) as a function of the amount of training data.

We show two illustrative experiments in figure 1a; please refer to Appendix D.1 for all simulations. We can see that *although the two $\mathcal{V}$-families used are misspecified with respect to the true underlying (conditional) distributions*, the estimated Chow-Liu trees are much more accurate across all data regimes, with **CPC** (blue) being the best alternative. Surprisingly, $\mathcal{V}$-information(Gaussian) works consistently well in all cases and only requires about 100 samples to recover the ground-truth Chow-Liu tree in simulation-A.

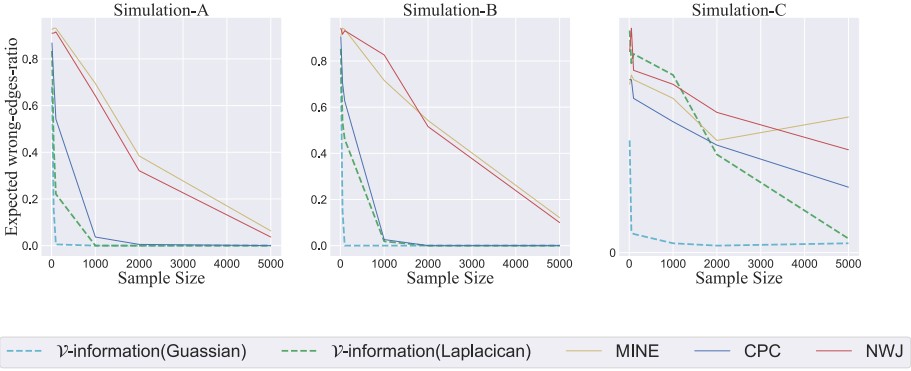

(a) Chow-Liu tree Construction

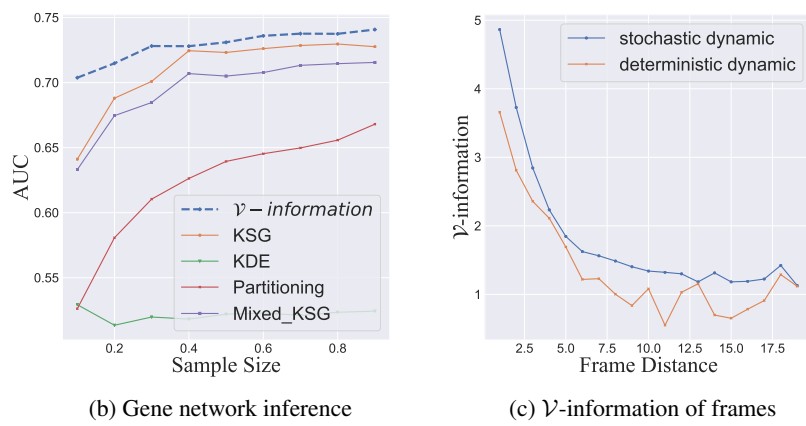

(b) Gene network inference

(c) $\mathcal{V}$-information of frames

Figure 1: (a) The expected wrong-edges-ratio of algorithm 1 with different $\mathcal{V}$ and other mutual information estimators-based algorithms from sample size 10 to $5 \times 10^3$. (b) AUC curve for gene regulatory network inference. (c) The predictive $\mathcal{V}$-information versus frame distance.

## 6.2 GENE REGULATORY NETWORK INFERENCE

Mutual information between pairs of gene expressions is often used to construct gene regulatory networks. We evaluate $\mathcal{V}$-information on the in-silico dataset from the DREAM5 challenge (Marbach et al., 2012) and use the setup of Gao et al. (2017), where 20 genes with 660 datapoints are utilized to evaluate all methods. We compare with state-of-the-art non-parametric Shannon mutual information estimators in this low dimensional setting: **KDE**, the traditional kernel density estimator; the **KSG** estimator (Kraskov et al., 2004); the **Mixed KSG** estimator (Gao et al., 2017) and **Partitioning**, an adaptive partitioning estimator (Darbellay & Vajda, 1999) implemented by Szabó (2014). For fair comparison with these low dimensional estimators, we select $\mathcal{V} = \{f : f[x] = \mathcal{N}(g(x), \frac{1}{2}), x \in \mathcal{X}; f[\varnothing] = \mathcal{N}(\mu, \frac{1}{2}) | \mu \in \text{range}(g)\}$, where $g$ is a 3-rd order polynomial.

The task is to predict whether a directed edge between genes exists in the ground-truth gene network. We use the estimated mutual information and $\mathcal{V}$-information for gene pairs as the test statistic to obtain the AUC for various methods. As shown in Figure 1b, our method outperforms all other methods in network inference under different fractions of data used for estimation. The natural information measure in this task is asymmetry since the goal is to find the pairs of genes $(A_i, B_i)$s in which $A_i$ regulates $B_i$, thus $\mathcal{V}$-information is more suitable for such case than mutual information.

### 6.3 RECOVERING THE ORDER OF VIDEO FRAMES

Let $X_1, \cdots, X_{20}$ be random variables each representing a frame in videos from the Moving-MNIST dataset, which contains 10,000 sequences each of length 20 showing two digits moving with stochastic dynamics. Can Algorithm 1 be used to recover the natural (causal) order of the frames? Intuitively, predictability should be inversely related with frame distance, thus enabling structure learning. Using a conditional PixelCNN++ (Salimans et al., 2017) as predictive family $\mathcal{V}$, we shown in Figure 1c that predictive $\mathcal{V}$-information does indeed decrease with frame distance, despite some fluctuations when the frame distances are large. Using Algorithm 1 to construct a Chow-Liu tree, we find that *the tree perfectly recovers the relative order of the frames*.

We also generate a Deterministic-Moving-MNIST dataset, where digits move according to *deterministic* dynamics. From the perspective of Shannon mutual information, every pair of frames has the same mutual information. Hence, standard Chow-Liu tree learning algorithm would fail to discover the natural ordering of the frames (causal structure). In contrast, once we constrain the observer to PixelCNN++ models, algorithm 1 with predictive $\mathcal{V}$-information can still recover the order of different frames when the frame distances are relatively small (less than 9). Compared to the stochastic dynamics case, $\mathcal{V}$-information is more irregular with increasing frame distance, since the PixelCNN++ tends to overfit.

### 6.4 INFORMATION THEORETIC APPROACHES TO FAIRNESS

The goal of fair representation learning is to map inputs $X \in \mathcal{X}$ to a feature space $Z \in \mathcal{Z}$ such that the mutual information between $Z$ and some sensitive attribute $U \in \mathcal{U}$ (such as race or gender) is minimized. The motivation is that using $Z$ (instead of $X$) as input we can no longer use the sensitive attributes $U$ to make decisions, thus ensuring some notion of fairness. Existing methods obtain fair representations by optimizing against an "adversarial" discriminator so that the discriminator cannot predict $U$ from $Z$ (Edwards & Storkey, 2015; Louizos et al., 2015; Madras et al., 2018; Song et al., 2018). Under some assumptions on $U$ and $\mathcal{V}$, we show in Appendix D.2 that these works actually use $\mathcal{V}$-information minimization as part of their objective, where $\mathcal{V}$ depends on the functional form of the discriminator.

However, it is clear from the $\mathcal{V}$-information perspective that features trained with $\mathcal{V}_A$-information minimization might not generalize to $\mathcal{V}_B$-information and vice versa. To illustrate this, we use a function family $\mathcal{V}_j$ as the attacker to extract information from features trained with $I_{\mathcal{V}_i}(Z \to U)$ minimization, where all the $\mathcal{V}$s are neural nets. On three datasets commonly used in the fairness literature (Adult, German, Heritage), previous methods work well at preventing information "leak" against the class of adversary they've been trained on, but fail when we consider different ones. As shown in Figure 3b in Appendix, the diagonal elements in the matrix are usually the smallest in rows, indicating that the attacker function family $\mathcal{V}_i$ extracts more information on featured trained with $\mathcal{V}_{j(j \neq i)}$-information minimization. This challenges the generalizability of fair representations in previous works. Please refer to Appendix D.2 for details.

## 7 RELATED WORK

**Alternative definitions of Information** Several alternative definitions of mutual information are available in the literature. Renyi entropy and Renyi mutual information (Lenzi et al., 2000) extend Shannon information by replacing KL divergence with $f$-divergences. However, they have the same difficulty when applied to high dimensional problems as Shannon information.

The line of work most related to ours is the $H$ entropy and $H$ mutual information (DeGroot et al., 1962; Grünwald et al., 2004), which associate a definition of entropy to every prediction loss. However, there are two key differences. First, literatures in $H$ entropy only consider a few special types of prediction functions that serve unique theoretical purposes; for example, (Duchi et al., 2018) considers the set of all functions on a feature space to prove surrogate risk consistency, and (Grünwald et al., 2004) only considers the $H$ entropy to prove the duality between maximum entropy and worst-case loss minimization. In contrast, our definition takes a completely different perspective — emphasizing bounded computation and intuitive properties of "usable" information. Furthermore $H$ entropy still suffers from difficulty of estimation in high dimension because the definitions do not restrict to functions with small complexity (e.g. Rademacher complexity).

**Mutual information estimation** The estimation of mutual information in the machine learning field is often on the continuous underlying distribution. For non-parametric mutual information estimators, many methods have exploited the $3H$ principle to calculate the mutual information, such as the Kernel density estimator (Paninski & Yajima, 2008), k-Nearest-Neighbor estimator and the KSG estimator (Kraskov et al., 2004). However, these non-parametric estimators usually aren't scalable to high dimension. Recently, several works utilize the variational lower bounds of MI to design MI estimator based on deep neural network in order to estimate MI of high dimension continuous random variables (Nguyen et al., 2010; van den Oord et al., 2018; Belghazi et al., 2018).

## 8 CONCLUSION

We defined and investigated $\mathcal{V}$-information, a variational extension to classic mutual information that incorporates computational constraints. Unlike Shannon mutual information, $\mathcal{V}$-information attempts to capture usable information, and has very different properties, such as invalidating the data processing inequality. In addition, $\mathcal{V}$-information can be provably estimated, and can thus be more effective for structure learning and fair representation learning.

### ACKNOWLEDGEMENTS

This research was supported by AFOSR (FA9550-19-1-0024), NSF (#1651565, #1522054, #1733686), ONR, and FLI.

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

# A PROOFS

## A.1 PROOF OF PROPOSITION 1

**Proposition 1.** *For $\mathcal{V}$-entropy and $\mathcal{V}$-information, we have*

1. *Let $\Omega$ be as in Def. 1. Then $H_\Omega(Y)$ is the Shannon entropy, $H_\Omega(Y \mid X)$ is the Shannon conditional entropy, and $I_\Omega(Y \to X)$ is the Shannon mutual information.*

2. *Let $\mathcal{Y} = \mathbb{R}^d$ and $\mathcal{V} = \{f : \{\varnothing\} \to P_\mu \mid \mu \in \mathbb{R}^d\}$, where $P_\mu$ is the distribution with density $y \mapsto \frac{1}{Z} e^{-\|y-\mu\|_2}$ where $Z = \int e^{-\|y-\mu\|_2} dy$, then the $\mathcal{V}$-entropy of a random variable $Y$ equals its mean absolute deviation, up to an additive constant.*

3. *Let $\mathcal{Y} = \mathbb{R}^d$ and $\mathcal{V} = \{f : \{\varnothing\} \to \mathcal{N}(\mu, \Sigma) \mid \mu \in \mathbb{R}^d, \Sigma = 1/2 I_{d\times d}\}$, then the $\mathcal{V}$-entropy of a random variable $Y$ equals the trace of its covariance $\mathrm{tr}\,(\mathrm{Cov}(Y))$, up to an additive constant.*

4. *Let $\mathcal{V} = \{f : \{\varnothing\} \to Q_{\mathbf{t},\theta}, \theta \in \Theta\}$, where $Q_{\mathbf{t},\theta}$ is a distribution in a minimal exponential family with sufficient statistics $\mathbf{t} : \mathcal{Y} \to \mathbb{R}^d$ and set of natural parameters $\Theta$. For a random variable $Y$ with expected sufficient statistics $\mu_Y = \mathbb{E}[\mathbf{t}(Y)]$, the $\mathcal{V}$-entropy of $Y$ is the maximum Shannon entropy over all random variables $\hat{Y}$ with identical expected sufficient statistics, i.e. $\mathbb{E}[\mathbf{t}(\hat{Y})] = \mu_Y$.*

5. *Let $\mathcal{Y} = \mathbb{R}^d$, $\mathcal{X}$ be any vector space, and $\mathcal{V} = \{f : x \mapsto \mathcal{N}(\phi(x), \Sigma), x \in \mathcal{X}; \varnothing \mapsto \mathcal{N}(\mu, \Sigma) \mid \mu \in \mathbb{R}^d; \Sigma = 1/2 I_{d\times d}, \phi \in \Phi\}$, where $\Phi$ is the set of linear functions $\{\phi : \mathcal{X} \to \mathbb{R}^d\}$, then $\mathcal{V}$-information $I_\mathcal{V}(X \to Y)$ equals the (unnormalized) maximum coefficient of determination $R^2 \cdot \mathrm{tr}\,(\mathrm{Cov}(Y))$ for linear regression.*

*Proof.* (1)

Let $P_{Y|x}$ denote the density function of random variable $Y$ conditioned on $X = x$ (we denote this random variable as $Y \mid x$).

$$
\begin{aligned}
H_\Omega(Y|X) &= \inf_{f \in \Omega} \mathbb{E}_{x,y \sim X,Y}\left[\log \frac{1}{f[x](y)}\right] = \inf_{f \in \Omega} \mathbb{E}_{x \sim X}\mathbb{E}_{y \sim Y|x}\left[\log \frac{P_{Y|x}(y)}{f[x](y)P_{Y|x}(y)}\right] \\
&= \inf_{f \in \Omega} \mathbb{E}_{x \sim X}\left[\mathrm{KL}(P_{Y|x}\|f[x]) + H(Y|x)\right] \\
&= \mathbb{E}_{x \sim X}\left[H(Y|x)\right] = H(Y|X)
\end{aligned}
\tag{9}
$$

where infimum is achieved for $f$ where $f[x] = P_{Y|x}$ and $H$ is the Shannon (conditional) entropy. The same proof technique can be used to show that $H_\Omega(Y) = H(Y)$, with the infimum achieved by $f$ where $f[\varnothing] = P_Y$. Hence we have

$$
I_\Omega(Y \to X) = H_\Omega(Y) - H_\Omega(Y|X) = H(Y) - H(Y|X) = I(Y;X)
\tag{10}
$$

(2)

$$
\begin{aligned}
H_\mathcal{V}(Y) &= \inf_{f \in \mathcal{V}} \mathbb{E}_{y \sim Y}\left[-\log f[\varnothing](y)\right] = \inf_{\mu \in \mathbb{R}^d} \mathbb{E}_{y \sim Y}\left[-\log \frac{1}{Z} e^{-\|y-\mu\|_2}\right] \\
&= \inf_{\mu \in \mathbb{R}^d} \mathbb{E}_{y \sim Y}\left[\| y - \mu \|_2\right] + \log Z \\
&= \mathrm{MAD}(Y) + \log Z
\end{aligned}
\tag{11}
$$

where MAD denotes mean absolute deviation $\mathbb{E}_{y \sim Y}\left[\| y - \mathbb{E}[Y] \|_2\right]$.

(3)

$$H_{\mathcal{V}}(Y) = \inf_{f \in \mathcal{V}} \mathbb{E}_{y \sim Y} \left[ -\log f[\varnothing](y) \right]$$

$$= \inf_{\mu \in \mathbb{R}^d} \mathbb{E}_{y \sim Y} \left[ -\log \frac{1}{(2\pi)^{\frac{d}{2}} |\Sigma|^{\frac{1}{2}}} e^{-\frac{1}{2}(y-\mu)^T \Sigma^{-1}(y-\mu)} \right]$$

$$= \inf_{\mu \in \mathbb{R}^d} \mathbb{E}_{y \sim Y} [(y-\mu)^T(y-\mu)] + \frac{d}{2} \log \pi$$

$$= \inf_{\mu \in \mathbb{R}^d} \mathbb{E}_{y \sim Y} [\text{tr} \left((y-\mu)(y-\mu)^T\right)] + \frac{d}{2} \log \pi \qquad \text{(Cyclic property of trace)}$$

$$= \text{tr} \left(\text{Cov}(Y)\right) + \frac{d}{2} \log \pi \qquad \text{(Linearity of trace)}$$

(4) The density function of an exponential family distribution with sufficient statistics $\mathbf{t}$ is $y \mapsto \exp\left(\theta \cdot \mathbf{t}(y) - A(\theta)\right)$ where $A(\theta)$ is the partition function.

$$H_{\mathcal{V}}(Y) = \inf_{f \in \mathcal{V}} \mathbb{E}_{y \sim Y} \left[ -\log f[\varnothing](y) \right] = \inf_{\theta \in \Theta} \mathbb{E}_{y \sim Y} \left[ -\log \exp\left(\theta \cdot \mathbb{E}_{y \sim Y}[\mathbf{t}(y)] - A(\theta)\right) \right]$$

$$= -\sup_{\theta \in \Theta} \left(\theta \cdot \mathbb{E}_{y \sim Y}[\mathbf{t}(y)] - A(\theta)\right)]$$

$$= -A^*(\mathbb{E}_{y \sim Y}[\mathbf{t}(y)]) \qquad (12)$$

where $A^*$ is the Fenchel dual of the log-partition function $A(\theta)$. Under mild conditions (Wainwright et al., 2008)

$$-A^*(\mu) = H(P_\mu)$$

where $P_\mu$ is the maximum entropy distribution out of all distributions satisfying $\mathbb{E}_{y \sim P_\mu}[\mathbf{t}(y)] = \mu$ (Jaynes, 1982), and $H(\cdot)$ is the Shannon entropy.

(5) Assume random variable $Y \in \mathbb{R}^d$, $\mathcal{V} = \{f : x \mapsto \mathcal{N}(\phi(x), \Sigma), x \in \mathcal{X}; \varnothing \mapsto \mathcal{N}(\mu, \Sigma) | \mu \in \mathbb{R}^d; \Sigma = \frac{1}{2} I_{d \times d}; \phi \in \Phi\}$. Then the $\mathcal{V}$-information from $X$ to $Y$ is

$$I_{\mathcal{V}}(X \to Y) = H_{\mathcal{V}}(Y) - H_{\mathcal{V}}(Y|X)$$

$$= \inf_{\mu \in \mathbb{R}^d} \mathbb{E}_{y \sim Y} \left[ -\log \frac{1}{(2\pi)^{\frac{d}{2}} |\Sigma|^{\frac{1}{2}}} e^{-\|y-\mu\|_2^2} \right] - \inf_{\phi \in \Phi} \mathbb{E}_{x,y \sim X,Y} \left[ -\log \frac{1}{(2\pi)^{\frac{d_y}{2}} |\Sigma|^{\frac{1}{2}}} e^{-\|y-\phi(x)\|_2^2} \right]$$

$$= \inf_{\mu \in \mathbb{R}^d} \mathbb{E}_{x,y \sim X,Y} \left[ \| y - \mu \|_2^2 \right] - \inf_{\phi \in \Phi} \mathbb{E}_{x,y \sim X,Y} \left[ \| y - \phi(x) \|_2^2 \right]$$

$$= \text{tr} \left(\text{Cov}(Y)\right) \left( 1 - \frac{\inf_{\phi \in \Phi} \mathbb{E}_{x,y \sim X,Y} \left[ \| y - \phi(x) \|_2^2 \right]}{\text{tr} \left(\text{Cov}(Y)\right)} \right)$$

$$= \text{tr} \left(\text{Cov}(Y)\right) R^2 \qquad (13)$$

$\square$

## A.2 Proof of Proposition 2

**Proposition 2.** *Let $Y$ and $X$ be any random variables on $\mathcal{Y}$ and $\mathcal{X}$, and $\mathcal{V}$ and $\mathcal{U}$ be any predictive families, then we have*

1. **Monotonicity**: *If $\mathcal{V} \subseteq \mathcal{U}$, then $H_{\mathcal{V}}(Y) \geq H_{\mathcal{U}}(Y)$, $H_{\mathcal{V}}(Y \mid X) \geq H_{\mathcal{U}}(Y \mid X)$.*

2. **Non-Negativity**: *$I_{\mathcal{V}}(X \to Y) \geq 0$.*

3. **Independence**: *If $X$ is independent of $Y$, $I_{\mathcal{V}}(X \to Y) = I_{\mathcal{V}}(Y \to X) = 0$.*

*Proof.* (1)

$$H_{\mathcal{V}}(Y) = \inf_{f \in \mathcal{V}} \mathbb{E}_{y \sim Y} \left[ \log \frac{1}{f[\varnothing](y)} \right] \geq \inf_{f \in \mathcal{U}} \mathbb{E}_{y \sim Y} \left[ \log \frac{1}{f[\varnothing](y)} \right] = H_{\mathcal{U}}(Y) \qquad (14)$$

$$H_{\mathcal{V}}(Y|X) = \inf_{f \in \mathcal{V}} \mathbb{E}_{x,y \sim X,Y} \left[ \log \frac{1}{f[x](y)} \right] \geq \inf_{f \in \mathcal{U}} \mathbb{E}_{x,y \sim X,Y} \left[ \log \frac{1}{f[x](y)} \right] = H_{\mathcal{U}}(Y|X) \qquad (15)$$

The inequalities (14) and (15) are because we are taking the infimum over a larger set.

**(2)**

Denote $\mathcal{V}_\varnothing \subset \mathcal{V}$ as the subset of $f$ that satisfy $f[x] = f[\varnothing], \forall x \in \mathcal{X}$.

$$
\begin{aligned}
H_\mathcal{V}(Y) &= \inf_{f \in \mathcal{V}} \mathbb{E}_{x,y \sim X,Y} \left[ -\log f[\varnothing](y) \right] \\
&= \inf_{f \in \mathcal{V}_\varnothing} \mathbb{E}_{x,y \sim X,Y} \left[ -\log f[\varnothing](y) \right] \qquad \text{(By Optional Ignorance)} \\
&= \inf_{f \in \mathcal{V}_\varnothing} \mathbb{E}_{x,y \sim X,Y} \left[ -\log f[x](y) \right] \\
&\geq \inf_{f \in \mathcal{V}} \mathbb{E}_{x,y \sim X,Y} \left[ -\log f[x](y) \right] = H_\mathcal{V}(Y \mid X)
\end{aligned}
$$

Therefore

$$
I_\mathcal{V}(Y \to X) = H_\mathcal{V}(Y) - H_\mathcal{V}(Y|X) \geq 0
$$

**(3)**

Denote $\mathcal{V}_\varnothing \subset \mathcal{V}$ as the subset of $f$ that satisfy $f[x] = f[\varnothing], \forall x \in \mathcal{X}$.

$$
\begin{aligned}
H_\mathcal{V}(Y \mid X) &= \inf_{f \in \mathcal{V}} \mathbb{E}_{x,y \sim X,Y} \left[ -\log f[x](y) \right] \\
&= \inf_{f \in \mathcal{V}} \mathbb{E}_{x \sim X} \mathbb{E}_{y \sim Y} \left[ -\log f[x](y) \right] \qquad \text{(Independence)} \\
&\geq \mathbb{E}_{x \sim X} \left[ \inf_{f \in \mathcal{V}} \mathbb{E}_{y \sim Y} \left[ -\log f[x](y) \right] \right] \qquad \text{(Jensen)} \\
&= \mathbb{E}_{x \sim X} \left[ \inf_{f \in \mathcal{V}_\varnothing} \mathbb{E}_{y \sim Y} \left[ -\log f[x](y) \right] \right] \qquad \text{(Optional Ignorance)} \\
&= \inf_{f \in \mathcal{V}_\varnothing} \mathbb{E}_{y \sim Y} \left[ -\log f[\varnothing](y) \right] \qquad \text{(No dependence on } x\text{)} \\
&\geq \inf_{f \in \mathcal{V}} \mathbb{E}_{y \sim Y} \left[ -\log f[\varnothing](y) \right] = H_\mathcal{V}(Y)
\end{aligned}
$$

Therefore $I_\mathcal{V}(Y \to X) = H_\mathcal{V}(Y) - H_\mathcal{V}(Y|X) \leq 0$. Combined with the Proposition 2.2 that $I_\mathcal{V}(X \to Y)$ must be non-negative, $I_\mathcal{V}(X \to Y)$ must be 0.

$\square$

### A.3 PROOF OF THEOREM 1

**Theorem 1.** *Assume $\forall f \in \mathcal{V}, x \in \mathcal{X}, y \in \mathcal{Y}, \log f[x](y) \in [-B, B]$. Then for any $\delta \in (0, 0.5)$, with probability at least $1 - 2\delta$, we have:*

$$
\left| I_\mathcal{V}(X \to Y) - \hat{I}_\mathcal{V}(X \to Y; \mathcal{D}) \right| \leq 4\mathfrak{R}_{|\mathcal{D}|}(\mathcal{G}_\mathcal{V}) + 2B\sqrt{\frac{2\log\frac{1}{\delta}}{|\mathcal{D}|}} \qquad (5)
$$

*where we define the function family $\mathcal{G}_\mathcal{V} = \{g | g(x, y) = \log f[x](y), f \in \mathcal{V}\}$, and $\mathfrak{R}_N(\mathcal{G})$ denotes the Rademacher complexity of $\mathcal{G}$ with sample number $N$.*

Before proving theorem 1, we introduce two lemmas. Proofs for these Lemmas follow the same strategy as theorem 8 in Bartlett & Mendelson (2001):

**Lemma 3.** *Let $X, Y$ be two random variables taking values in $\mathcal{X}, \mathcal{Y}$ and $\mathcal{D}$ denotes the set of samples drawn from the joint distribution over $\mathcal{X} \times \mathcal{Y}$. Assume $\forall f \in \mathcal{V}, x \in \mathcal{X}, y \in \mathcal{Y}, \log f[x](y) \in [-B, B]$. Take $\hat{f} = \arg\min_{f \in \mathcal{V}} \frac{1}{|\mathcal{D}|} \sum_{x_i, y_i \in \mathcal{D}} -\log f[x_i](y_i)$, then $\forall \delta \in (0, 1)$, with probability at least $1 - \delta$, we have:*

$$
\left| H_\mathcal{V}(Y|X) - \frac{1}{|\mathcal{D}|} \sum_{x_i, y_i \in \mathcal{D}} -\log \hat{f}[x_i](y_i) \right| \leq 2\mathfrak{R}_{|\mathcal{D}|}(\mathcal{G}_\mathcal{V}) + 2B\sqrt{\frac{2\log\frac{1}{\delta}}{|\mathcal{D}|}} \qquad (16)
$$

*Proof.* We apply McDiarmid's inequality to the function $\Phi$ defined for any sample $\mathcal{D}$ by

$$\Phi(\mathcal{D}) = \sup_{f \in \mathcal{V}} \left| \mathbb{E}_{x,y} \left[ -\log f[x](y) \right] - \frac{1}{|\mathcal{D}|} \sum_{x_i, y_i \in \mathcal{D}} -\log f[x_i](y_i) \right| \qquad (17)$$

Let $\mathcal{D}$ and $\mathcal{D}'$ be two samples differing by exactly one point, then since the difference of suprema does not exceed the supremum of the difference and $\forall f \in \mathcal{V}, x \in \mathcal{X}, y \in \mathcal{Y}, \log f[x](y) \in [-B, B]$, we have:

$$\Phi(\mathcal{D}) - \Phi(\mathcal{D}')$$

$$\leq \sup_{f \in \mathcal{V}} \left[ \left| \frac{1}{|\mathcal{D}|} \sum_{x_i, y_i \in \mathcal{D}} \log f[x_i](y_i) - \mathbb{E}_{x,y} \left[ \log f[x](y) \right] \right| - \left| \frac{1}{|\mathcal{D}'|} \sum_{x_i, y_i \in \mathcal{D}'} \log f[x_i](y_i) - \mathbb{E}_{x,y} \left[ \log f[x](y) \right] \right| \right]$$

$$\leq \sup_{f \in \mathcal{V}} \left| \frac{1}{|\mathcal{D}|} \sum_{x_i, y_i \in \mathcal{D}} -\log f[x_i](y_i)| - \frac{1}{|\mathcal{D}'|} \sum_{x_i, y_i \in \mathcal{D}'} -\log f[x_i](y_i) \right|$$

$$\leq \frac{2B}{|\mathcal{D}|}$$

then by McDiarmid's inequality, for any $\delta \in (0, 1)$, with probability at least $1 - \delta$, the following holds:

$$\Phi(\mathcal{D}) \leq \mathbb{E}_{\mathcal{D}}[\Phi(\mathcal{D})] + B \sqrt{\frac{2 \log \frac{1}{\delta}}{|\mathcal{D}|}} \qquad (18)$$

Then we bound the $\mathbb{E}_{\mathcal{D}}[\Phi(\mathcal{D})]$ term:

$$\mathbb{E}_{\mathcal{D}}[\Phi(\mathcal{D})] = \mathbb{E}_{\mathcal{D}} \left[ \sup_{f \in \mathcal{V}} \left| \mathbb{E}_{x,y} \left[ -\log f[x](y) \right] - \frac{1}{|\mathcal{D}|} \sum_{x_i, y_i \in \mathcal{D}} -\log f[x_i](y_i) \right| \right] \qquad (19)$$

$$= \mathbb{E}_{\mathcal{D}} \left[ \sup_{f \in \mathcal{V}} \left| \mathbb{E}_{\mathcal{D}'} \left[ \frac{1}{|\mathcal{D}'|} \sum_{x_i', y_i' \in \mathcal{D}'} \log f[x_i'](y_i') \right] - \frac{1}{|\mathcal{D}|} \sum_{x_i, y_i \in \mathcal{D}} \log f[x_i](y_i) \right| \right] \qquad (20)$$

$$\leq \mathbb{E}_{\mathcal{D}} \left[ \sup_{f \in \mathcal{V}} \mathbb{E}_{\mathcal{D}'} \left| \frac{1}{|\mathcal{D}'|} \sum_{x_i', y_i' \in \mathcal{D}'} \log f[x_i'](y_i')| - \frac{1}{|\mathcal{D}|} \sum_{x_i, y_i \in \mathcal{D}} \log f[x_i](y_i) \right| \right] \qquad (21)$$

$$\leq \mathbb{E}_{\mathcal{D}, \mathcal{D}'} \left[ \sup_{f \in \mathcal{V}} \left| \frac{1}{|\mathcal{D}'|} \sum_{x_i', y_i' \in \mathcal{D}'} \log f[x_i'](y_i')| - \frac{1}{|\mathcal{D}|} \sum_{x_i, y_i \in \mathcal{D}} \log f[x_i](y_i) \right| \right] \qquad (22)$$

$$= \mathbb{E}_{\mathcal{D}, \mathcal{D}'} \left[ \sup_{f \in \mathcal{V}} \left| \frac{1}{|\mathcal{D}|} \sum_{i=1}^{|\mathcal{D}|} (\log f[x_i'](y_i') - \log f[x_i](y_i)) \right| \right] \qquad (23)$$

$$\leq \mathbb{E}_{\mathcal{D}, \mathcal{D}', \sigma} \left[ \sup_{f \in \mathcal{V}} \left| \frac{1}{|\mathcal{D}|} \sum_{i=1}^{|\mathcal{D}|} \sigma_i (\log f[x_i'](y_i') - \log f[x_i](y_i)) \right| \right] \qquad (24)$$

$$\leq \mathbb{E}_{\mathcal{D}, \sigma} \left[ \sup_{f \in \mathcal{V}} \left| \frac{1}{|\mathcal{D}|} \sum_{i=1}^{|\mathcal{D}|} \sigma_i \log f[x_i](y_i) \right| \right] + \mathbb{E}_{\mathcal{D}', \sigma} \left[ \sup_{f \in \mathcal{V}} \left| \frac{1}{|\mathcal{D}|} \sum_{i=1}^{|\mathcal{D}|} \sigma_i \log f[x_i'](y_i') \right| \right] \qquad (25)$$

$$= 2 \mathbb{E}_{\mathcal{D}, \sigma} \left[ \sup_{f \in \mathcal{V}} \left| \frac{1}{|\mathcal{D}|} \sum_{i=1}^{|\mathcal{D}|} \sigma_i \log f[x_i](y_i) \right| \right] \qquad (26)$$

$$= 2\mathbb{E}_{\mathcal{D},\sigma}\left[\sup_{g\in\mathcal{G}}\left|\frac{1}{|\mathcal{D}|}\sum_{i=1}^{|\mathcal{D}|}\sigma_i g(x_i,y_i)\right|\right] = 2\mathfrak{R}_{|\mathcal{D}|}(\mathcal{G}_\mathcal{V}) \tag{27}$$

where $\sigma_i$s are Rademacher variables that is uniform in $\{-1,+1\}$. Inequality (22) follows from the convexity of $\sup$, inequality (24) follows from the symmetrization argument for $\ell_1$ norm for Radermacher random variables (Ledoux & Talagrand (2013), Section 6.1), inequality (21) follows from the convexity of $|x-c|$. (27) follows from the definition of $\mathcal{G}$ and Rademacher complexity.

Finally, combining inequality (18) and (27) yields for all $f\in\mathcal{V}$, with probability at least $1-\delta$

$$\left|\mathbb{E}_{x,y}[-\log f[x](y)] - \frac{1}{|\mathcal{D}|}\sum_{x_i,y_i\in\mathcal{D}} -\log f[x_i](y_i)\right| \le 2\mathfrak{R}_{|\mathcal{D}|}(\mathcal{G}_\mathcal{V}) + B\sqrt{\frac{2\log\frac{1}{\delta}}{|\mathcal{D}|}} \tag{28}$$

In particular, the inequality holds for $\hat{f} = \arg\min_{f\in\mathcal{V}}\frac{1}{|\mathcal{D}|}\sum_{x_i,y_i\in\mathcal{D}} -\log f[x_i](y_i)$ and $\tilde{f} = \arg\min_{f\in\mathcal{V}}\mathbb{E}_{x,y\sim X,Y}[-\log f[x](y)]$. Then we have:

$$\mathbb{E}_{x,y\sim X,Y}\left[-\log\tilde{f}[x](y)\right] - \frac{1}{|\mathcal{D}|}\sum_{x_i,y_i\in\mathcal{D}} -\log\tilde{f}[x_i](y_i) \le H_\mathcal{V}(Y|X) - \frac{1}{|\mathcal{D}|}\sum_{x_i,y_i\in\mathcal{D}} -\log\hat{f}[x_i](y_i)$$

$$\le \mathbb{E}_{x,y\sim X,Y}\left[-\log\hat{f}[x](y)\right] - \frac{1}{|\mathcal{D}|}\sum_{x_i,y_i\in\mathcal{D}} -\log\hat{f}[x_i](y_i)$$

Hence the bound (16) holds. $\qquad\square$

Similar bounds can be derived for $H_\mathcal{V}(Y)$ when we choose the domain of $x$ to be $\mathcal{X} = \{\varnothing\}$:

**Lemma 4.** *Let $Y$ be random variable taking values in $\mathcal{Y}$ and $\mathcal{D}$ denotes the set of samples drawn from the underlying distribution $P(Y)$. Assume $\forall f\in\mathcal{V}, y\in\mathcal{Y}, \log f[\varnothing](y)\in[-B,B]$. Take $\hat{f} = \arg\min_{f\in\mathcal{V}}\frac{1}{|\mathcal{D}|}\sum_{x_i,y_i\in\mathcal{D}} -\log f[\varnothing](y_i)$, then for any $\delta\in(0,1)$, with probability at least $1-\delta$, we have:*

$$\left|H_\mathcal{V}(Y) - \frac{1}{|\mathcal{D}|}\sum_{y_i\in\mathcal{D}} -\log\hat{f}[\varnothing](y_i)\right| \le 2\mathfrak{R}_{|\mathcal{D}|}(\mathcal{G}_{\mathcal{V}^\varnothing}) + B\sqrt{\frac{2\log\frac{1}{\delta}}{|\mathcal{D}|}} \tag{29}$$

$$\le 2\mathfrak{R}_{|\mathcal{D}|}(\mathcal{G}_\mathcal{V}) + B\sqrt{\frac{2\log\frac{1}{\delta}}{|\mathcal{D}|}} \tag{30}$$

where $\mathcal{G}_{\mathcal{V}^\varnothing} = \{g|g(y) = \log f[\varnothing](y), f\in\mathcal{V}\}$.

*Proof.* The first inequality (29) can be derived similarly as Lemma 3. Since $\mathcal{V}$ is a predictive family, hence there exits a function $h:\mathcal{V}\to\mathcal{V}$, such that $h(f) = f'$ and $\forall x\in\mathcal{X}, f'[x] = f[\varnothing]$.

$$\mathfrak{R}_{|\mathcal{D}|}(\mathcal{G}_{\mathcal{V}^\varnothing}) = \mathbb{E}_{\mathcal{D},\sigma}\left[\sup_{f\in\mathcal{V}}\left|\frac{1}{|\mathcal{D}|}\sum_{i=1}^{|\mathcal{D}|}\sigma_i\log f[\varnothing](y_i)\right|\right]$$

$$= \mathbb{E}_{\mathcal{D},\sigma}\left[\sup_{f\in\mathcal{V}}\left|\frac{1}{|\mathcal{D}|}\sum_{i=1}^{|\mathcal{D}|}\sigma_i\log h(f)[x_i](y_i)\right|\right]$$

$$\le \mathbb{E}_{\mathcal{D},\sigma}\left[\sup_{f\in\mathcal{V}}\left|\frac{1}{|\mathcal{D}|}\sum_{i=1}^{|\mathcal{D}|}\sigma_i\log f[x_i](y_i)\right|\right] \tag{31}$$

$$= \mathfrak{R}_{|\mathcal{D}|}(\mathcal{G}_\mathcal{V})$$

The inequality (31) holds because of $h(\mathcal{V})\subseteq\mathcal{V}$. $\qquad\square$

Now we prove theorem 1:

**Theorem 1.** Assume $\forall f \in \mathcal{V}, x \in \mathcal{X}, y \in \mathcal{Y}, \log f[x](y) \in [-B, B]$, for any $\delta \in (0, 0.5)$, with probability at least $1 - 2\delta$, we have:

$$\left| I_\mathcal{V}(X \to Y) - \hat{I}_\mathcal{V}(X \to Y; \mathcal{D}) \right| \leq 4\mathfrak{R}_{|\mathcal{D}|}(\mathcal{G}_\mathcal{V}) + 2B\sqrt{\frac{2\log\frac{1}{\delta}}{|\mathcal{D}|}}$$

*Proof.* Define $\hat{f} = \underset{f \in \mathcal{V}}{\arg\min} \sum_{x_i, y_i \in \mathcal{D}} -\log f[x_i](y_i)$ and $\hat{f}_\varnothing = \underset{f \in \mathcal{V}}{\arg\min} \sum_{y_i \in \mathcal{D}} -\log f[\varnothing](y_i)$. Using the triangular inequality we have:

$$\left| I_\mathcal{V}(X \to Y) - \hat{I}_\mathcal{V}(X \to Y; \mathcal{D}) \right|$$

$$= \left| (H_\mathcal{V}(Y) - H_\mathcal{V}(Y|X)) - \left( \frac{1}{|\mathcal{D}|}\sum_{y_i \in \mathcal{D}} -\log \hat{f}_\varnothing[\varnothing](y_i) - \frac{1}{|\mathcal{D}|}\sum_{x_i, y_i \in \mathcal{D}} -\log \hat{f}[x_i](y_i) \right) \right|$$

$$\leq \left| \left( H_\mathcal{V}(Y) - \frac{1}{|\mathcal{D}|}\sum_{y_i \in \mathcal{D}} -\log \hat{f}_\varnothing[\varnothing](y_i) \right) - \left( H_\mathcal{V}(Y|X) - \frac{1}{|\mathcal{D}|}\sum_{x_i, y_i \in \mathcal{D}} -\log \hat{f}[x_i](y_i) \right) \right|$$

$$\leq \left| H_\mathcal{V}(Y|X) - \frac{1}{|\mathcal{D}|}\sum_{x_i, y_i \in \mathcal{D}} -\log \hat{f}[x_i](y_i) \right| + \left| H_\mathcal{V}(Y) - \frac{1}{|\mathcal{D}|}\sum_{y_i \in \mathcal{D}} -\log \hat{f}_\varnothing[\varnothing](y_i) \right| \quad (32)$$

For simplicity let

$$D_{Y|X} = \left| H_\mathcal{V}(Y|X) - \frac{1}{|\mathcal{D}|}\sum_{x_i, y_i \in \mathcal{D}} -\log \hat{f}[x_i](y_i) \right|$$

and

$$D_Y = \left| H_\mathcal{V}(Y) - \frac{1}{|\mathcal{D}|}\sum_{y_i \in \mathcal{D}} -\log \hat{f}_\varnothing[\varnothing](y_i) \right|$$

With inequality (32), Lemma 3 and Lemma 4, we have:

$$\Pr\left( \left| I_\mathcal{V}(X \to Y) - \hat{I}_\mathcal{V}(X \to Y; \mathcal{D}) \right| > 4\mathfrak{R}_{|\mathcal{D}|}(\mathcal{G}_\mathcal{V}) + 2B\sqrt{\frac{2\log\frac{1}{\delta}}{|\mathcal{D}|}} \right)$$

$$\leq \Pr\left( D_{Y|X} + D_Y > 4\mathfrak{R}_{|\mathcal{D}|}(\mathcal{G}_\mathcal{V}) + 2B\sqrt{\frac{2\log\frac{1}{\delta}}{|\mathcal{D}|}} \right) \quad \text{(Inequality (32))}$$

$$\leq \Pr\left( \left( D_{Y|X} > 2\mathfrak{R}_{|\mathcal{D}|}(\mathcal{G}_\mathcal{V}) + B\sqrt{\frac{2\log\frac{1}{\delta}}{|\mathcal{D}|}} \right) \vee \left( D_Y > 2\mathfrak{R}_{|\mathcal{D}|}(\mathcal{G}_\mathcal{V}) + B\sqrt{\frac{2\log\frac{1}{\delta}}{|\mathcal{D}|}} \right) \right)$$

$$\leq \Pr\left( D_{Y|X} > 2\mathfrak{R}_{|\mathcal{D}|}(\mathcal{G}_\mathcal{V}) + B\sqrt{\frac{2\log\frac{1}{\delta}}{|\mathcal{D}|}} \right) + \Pr\left( D_Y > 2\mathfrak{R}_{|\mathcal{D}|}(\mathcal{G}_\mathcal{V}) + B\sqrt{\frac{2\log\frac{1}{\delta}}{|\mathcal{D}|}} \right)$$

$$\text{(Union bound)}$$

$$\leq 2\delta \quad \text{(Lemma 3 and Lemma 4)}$$

Hence we have:

$$\Pr\left( \left| I_\mathcal{V}(X \to Y) - \hat{I}_\mathcal{V}(X \to Y; \mathcal{D}) \right| \leq 4\mathfrak{R}_{|\mathcal{D}|}(\mathcal{G}_\mathcal{V}) + 2B\sqrt{\frac{2\log\frac{1}{\delta}}{|\mathcal{D}|}} \right) \geq 1 - 2\delta$$

which completes the proof. $\square$

## A.4 Proof of Corollary 1.1

**Corollary 1.1.** *Assume* $\mathcal{X} = \{x \in \mathbb{R}^{d_x}, \|x\|_2 \le k_x\}$ *and* $\mathcal{Y} = \{y \in \mathbb{R}^{d_y}, \|y\|_2 \le k_y\}$. *If*

$$\mathcal{V} = \{f : f[x] = \mathcal{N}(Wx + b, I), f[\varnothing] = \mathcal{N}(c, I), W \in \mathbb{R}^{d_y \times d_x}, b, c \in \mathbb{R}^{d_y}, \|(W, b)\|_2 \le 1\}$$

*Denote* $M = (k_x + k_y)^2 + \log 2\pi$, *then* $\forall \delta \in (0, 0.5)$, *with probability at least* $1 - 2\delta$:

$$\left| I_{\mathcal{V}}(X \to Y) - \hat{I}_{\mathcal{V}}(X \to Y; \mathcal{D}) \right| \le \frac{M}{\sqrt{4|\mathcal{D}|}} \left( 1 + 4\sqrt{2\log\frac{1}{\delta}} \right)$$

The proof is an adaptation of the proof for theorem 3 in Kakade et al. (2008).

*Proof.* From theorem 1 we have:

$$\left| I_{\mathcal{V}}(X \to Y) - \hat{I}_{\mathcal{V}}(X \to Y; \mathcal{D}) \right| \le 4\mathfrak{R}_{|\mathcal{D}|}(\mathcal{G}_{\mathcal{V}}) + 2B\sqrt{\frac{2\log\frac{1}{\delta}}{|\mathcal{D}|}}$$

In the following $\|(W, b)\|_2$ is the matrix 2-norm of $(W, b)$, then the Rademacher term can be bounded as follows:

$$\mathfrak{R}_{|\mathcal{D}|}(\mathcal{G}_{\mathcal{V}}) = \frac{1}{|\mathcal{D}|} \mathbb{E}_\sigma \left[ \sup_{W, b, \|(W, b)\|_2 \le 1} \left| \sum_{i=1}^{|\mathcal{D}|} \sigma_i \left( \log \frac{1}{\sqrt{2\pi}} - \frac{1}{2}\|y_i - Wx_i - b\|_2^2 \right) \right| \right]$$

$$\le \frac{1}{|\mathcal{D}|} \mathbb{E}_\sigma \left[ \sup_{W, b, \|(W, b)\|_2 \le 1} \left| \sum_{i=1}^{|\mathcal{D}|} \sigma_i \left( -\frac{1}{2}\|y_i - Wx_i - b\|_2^2 \right) \right| \right] + \frac{1}{|\mathcal{D}|} \mathbb{E}_\sigma \left[ \left| \sum_{i=1}^{|\mathcal{D}|} \sigma_i \log \frac{1}{\sqrt{2\pi}} \right| \right] \tag{33}$$

The second term in RHS can be bounded as follows:

$$\frac{1}{|\mathcal{D}|} \mathbb{E}_\sigma \left[ \left| \sum_{i=1}^{|\mathcal{D}|} \sigma_i \log \frac{1}{\sqrt{2\pi}} \right| \right] \le \frac{1}{|\mathcal{D}|} \sqrt{\mathbb{E}_\sigma \left[ \left( \sum_{i=1}^{|\mathcal{D}|} \sigma_i \log \frac{1}{\sqrt{2\pi}} \right)^2 \right]} \qquad \text{(concavity of } x^{\frac{1}{2}}\text{)}$$

$$= \frac{1}{|\mathcal{D}|} \sqrt{|\mathcal{D}| * (\log \frac{1}{\sqrt{2\pi}})^2} \qquad \text{(Independence of } \sigma_i\text{s)}$$

$$= \sqrt{\frac{(\log \frac{1}{\sqrt{2\pi}})^2}{|\mathcal{D}|}} \tag{34}$$

The first term in RHS can be bounded as follows:

$$\frac{1}{|\mathcal{D}|} \mathbb{E}_{\mathcal{D}, \sigma} \left[ \sup_{W, b, \|(W, b)\|_2 \le 1} \left| \sum_{i=1}^{|\mathcal{D}|} \sigma_i \left( -\frac{1}{2}\|y_i - Wx_i - b\|^2 \right) \right| \right]$$

$$= \frac{1}{2|\mathcal{D}|} \mathbb{E}_{\mathcal{D}, \sigma} \left[ \sup_{W, b, \|(W, b)\|_2 \le 1} \left| \sum_{i=1}^{|\mathcal{D}|} \sigma_i \left( \|y_i - Wx_i - b\|^2 \right) \right| \right]$$

$$\le \frac{\max_i \|y_i\|_2^2}{2} \sqrt{\frac{1}{|\mathcal{D}|}} + \max_i \|x_i\|_2 \sqrt{\frac{\max_i \|y_i\|^2}{|\mathcal{D}|}}$$

$$+ \frac{1}{2|\mathcal{D}|} \mathbb{E}_{\mathcal{D}, \sigma} \left[ \sup_{W, b, \|(W, b)\|_2 \le 1} \left| \sum_{i=1}^{|\mathcal{D}|} \sigma_i \left( \|Wx_i + b\|^2 \right) \right| \right] \tag{35}$$

$$\leq \frac{\max_i \|y_i\|_2^2}{2} \sqrt{\frac{1}{|\mathcal{D}|}} + \max_i \|x_i\|_2 \sqrt{\frac{\max_i \|y_i\|^2}{|\mathcal{D}|}}$$

$$+ \frac{\max_i \|x_i\|_2}{2|\mathcal{D}|} \mathbb{E}_{\mathcal{D},\sigma} \left[ \sup_{W,b,\|(W,b)\|_2 \leq 1} \left| \sum_{i=1}^{|\mathcal{D}|} \sigma_i \left( \|W x_i + b\| \right) \right| \right] \tag{36}$$

$$\leq \frac{\max_i \|y_i\|_2^2}{2} \sqrt{\frac{1}{|\mathcal{D}|}} + \max_i \|x_i\|_2 \sqrt{\frac{\max_i \|y_i\|_2^2}{|\mathcal{D}|}} + \frac{\max_i \|x_i\|_2}{2} \sqrt{\frac{\max_i \|x_i\|_2^2}{|\mathcal{D}|}} \tag{37}$$

$$\leq \frac{M}{\sqrt{4|\mathcal{D}|}}$$

The inequalities (36) and (35) follow the same proof in (34).

Hence we have:

$$\mathfrak{R}_{|\mathcal{D}|}(\mathcal{G}_{\mathcal{V}}) \leq \frac{M}{\sqrt{4|\mathcal{D}|}} \tag{38}$$

In this example, we can bound the upper bound of functions $g \in \mathcal{G}_{\mathcal{V}}$ by

$$B = \sup_{x \in \mathcal{X}, y \in \mathcal{Y}, \|(W,b)\|_2 \leq 1} \left| \left( \log \frac{1}{\sqrt{2\pi}} - \frac{1}{2} \|y - W x - b\|_2^2 \right) \right|$$

$$\leq \sup_{x \in \mathcal{X}, y \in \mathcal{Y}, \|(W,b)\|_2 \leq 1} \log \frac{1}{\sqrt{2\pi}} + \frac{1}{2} \left( \|y\|_2^2 + \|W x + b\|_2^2 + 2\|y\| \|W x + b\| \right)$$

$$\leq \log \frac{1}{\sqrt{2\pi}} + \frac{1}{2}(k_x + k_y)^2 < M$$

Combining inequality (38) we arrive at the theorem. $\qquad \square$

## A.5   PROOF OF THEOREM 2

**Theorem 2.** *Let $\{X_i\}_{i=1}^m$ be the set of m random variables, $\mathcal{D}_{i,j}$ (resp. $\mathcal{D}_j$) be the set of samples drawn from $P(X_i, X_j)$ (resp. $P(X_j)$). Denote the optimal directed tree with maximum expected edge weights sum $C(g)$ as $g^*$ and the optimal directed tree constructed on the dataset $\mathcal{D}$ as $\hat{g}$. Then with the assumption in theorem 1, for any $\delta \in (0, \frac{1}{2m(m-1)})$, with probability at least $1 - 2m(m-1)\delta$, we have:*

$$C(\hat{g}) \geq C(g^*) - 2(m-1) \max_{i,j} \left\{ 2\mathfrak{R}_{\mathcal{D}_{i,j}}(\mathcal{G}_{\mathcal{V}_{i,j}}) + 2\mathfrak{R}_{\mathcal{D}_j}(\mathcal{G}_{\mathcal{V}_j}) + B\sqrt{2\log\frac{1}{\delta}}(|\mathcal{D}_j|^{-\frac{1}{2}} + |\mathcal{D}_{i,j}|^{-\frac{1}{2}}) \right\} \tag{8}$$

*Proof.* Let $C_{\mathcal{D}}(g^*)$ be the estimated sum of edge weights on dataset $\mathcal{D}$ of the tree $g^*$, i.e.,

$$C(g^*) = \sum_{i=2}^m \hat{I}_{\mathcal{V}_{t(g^*)(i),i}}(X_{t(g)(i)} \to X_i; \mathcal{D}).$$

where $t(g) : \mathbb{N} \to \mathbb{N}$ is the function mapping each non-root node of directed tree $g$ to its parent. The same notation for tree $\hat{g}$. Let

$$\epsilon = \max_{i,j} \left\{ \left| I_{\mathcal{V}}(X_i \to X_j) - \hat{I}_{\mathcal{V}}(X_i \to X_j; \mathcal{D}) \right| \right\}$$

be the maximum absolute estimation error of single edge weight. By the definition of $\epsilon$ we have $\forall g, |C(\hat{g}) - C_D(\hat{g})| \leq (m-1)\epsilon$, then:

$$C(\hat{g}) + (m-1)\epsilon \geq C_{\mathcal{D}}(\hat{g}) \geq C_{\mathcal{D}}(g^*) \geq C(g^*) - (m-1)\epsilon \tag{39}$$

From lemma 4 and lemma 3 we have:

$$\Pr\left(\epsilon > \max_{i,j}\left\{2\mathfrak{R}_{\mathcal{D}_{i,j}}(\mathcal{G}_{i,j}) + 2\mathfrak{R}_{\mathcal{D}_j}(\mathcal{G}_j) + B\sqrt{2\log\frac{1}{\delta}}(|\mathcal{D}_j|^{-\frac{1}{2}} + |\mathcal{D}_{i,j}|^{-\frac{1}{2}})\right\}\right)$$

$$\leq \Pr\left(\exists i,j, \left|I_{\mathcal{V}_{i,j}}(X_i \to X_j) - \hat{I}_{\mathcal{V}_{i,j}}(X_i \to X_j; \mathcal{D})\right| > 2\mathfrak{R}_{\mathcal{D}_{i,j}}(\mathcal{G}_{i,j}) + 2\mathfrak{R}_{\mathcal{D}_j}(\mathcal{G}_j) + B\sqrt{2\log\frac{1}{\delta}}(|\mathcal{D}_j|^{-\frac{1}{2}} + |\mathcal{D}_{i,j}|^{-\frac{1}{2}})\right)$$

$$\leq \Pr\left(\exists i,j, \left|\left(H_{\mathcal{V}_j}(X_j) - \frac{1}{|\mathcal{D}_j|}\sum_{x_j \in \mathcal{D}_j} -\log\hat{f}_\varnothing[\varnothing](x_j)\right) - \left(H_{\mathcal{V}_{i,j}}(X_j|X_i) - \frac{1}{|\mathcal{D}_{i,j}|}\sum_{x_i,x_j \in \mathcal{D}_{i,j}} -\log\hat{f}[x_i](x_j)\right)\right|\right.$$

$$\left. > 2\mathfrak{R}_{\mathcal{D}_{i,j}}(\mathcal{G}_{i,j}) + 2\mathfrak{R}_{\mathcal{D}_j}(\mathcal{G}_j) + B\sqrt{2\log\frac{1}{\delta}}(|\mathcal{D}_j|^{-\frac{1}{2}} + |\mathcal{D}_{i,j}|^{-\frac{1}{2}})\right)$$

$$\leq \Pr\left(\exists i,j, \left(\left|H_{\mathcal{V}_j}(X_j) - \frac{1}{|\mathcal{D}_j|}\sum_{x_j \in \mathcal{D}_j} -\log\hat{f}_\varnothing[\varnothing](x_j)\right| > 2\mathfrak{R}_{\mathcal{D}_j}(\mathcal{G}_j) + B\sqrt{2\log\frac{1}{\delta}}|\mathcal{D}_j|^{-\frac{1}{2}}\right)\right.$$

$$\left. \vee \left(\left|H_{\mathcal{V}_{i,j}}(X_j|X_i) - \frac{1}{|\mathcal{D}_{i,j}|}\sum_{x_i,x_j \in \mathcal{D}_{i,j}} -\log\hat{f}[x_i](x_j)\right| > 2\mathfrak{R}_{\mathcal{D}_{i,j}}(\mathcal{G}_{i,j}) + B\sqrt{2\log\frac{1}{\delta}}|\mathcal{D}_{i,j}|^{-\frac{1}{2}}\right)\right)$$

$$\leq m(m-1)2\delta \qquad\qquad\text{(By lemma 3, 4 and union bound)}$$

Hence

$$\Pr\left(\epsilon \leq \max_{i,j}\left\{2\mathfrak{R}_{\mathcal{D}_{i,j}}(\mathcal{G}_{i,j}) + 2\mathfrak{R}_{\mathcal{D}_j}(\mathcal{G}_j) + B\sqrt{2\log\frac{1}{\delta}}(|\mathcal{D}_j|^{-\frac{1}{2}} + |\mathcal{D}_{i,j}|^{-\frac{1}{2}})\right\}\right) \geq 1 - m(m-1)2\delta$$

$$(40)$$

Then combining inequality (39) and (40) we arrive at the result. $\qquad\square$

## B  ANALYSIS OF APPROXIMATE ESTIMATORS FOR SHANNON INFORMATION

We consider two approximate estimators for Shannon information. The first is the CPC (or InfoNCE in Poole et al. (2019)) estimator ($I_{\text{CPC}}$) proposed by van den Oord et al. (2018):

$$I_{\text{CPC}} = \mathbb{E}\left[\frac{1}{N}\sum_{i=1}^{N}\log\frac{f_\theta(x_i, y_i)}{\frac{1}{N}\sum_{j=1}^{N}f_\theta(x_i, y_j)}\right] \leq I(X; Y) \qquad (41)$$

where the expectation is over N independent samples form the joint distribution $\prod_i p(x_i, y_i)$.

The second is the NWJ estimator ($I_{\text{NWJ}}$) proposed by Nguyen et al. (2010):

$$I_{\text{NWJ}} = \mathbb{E}_{x,y\sim p(x,y)}\left[f_\theta(x,y)\right] - e^{-1}\mathbb{E}_{x,y\sim p(x)p(y)}\left[e^{f_\theta(x,y)}\right] \leq I(X; Y) \qquad (42)$$

In both cases, $f_\theta$ is a parameterized function, and the objectives are to maximize these lower bounds parameterized by $\theta$ to approximate mutual information. Ideally, with sufficiently flexible models and data, we would be able recover the true mutual information. However, these ideal cases does not carry over to practical scenarios.

For $I_{\text{CPC}}$, van den Oord et al. (2018) show that $I_{\text{CPC}}$ is no larger than $\log N$, where $N$ is the batch size. This means that the $I_{\text{CPC}}$ estimator will incur large bias when $I(X; Y) \geq \log N$. We provide a proof for completeness as follows.

**Proposition 3.** $\forall f_\theta : \mathcal{X} \times \mathcal{Y} \to \mathbb{R}^+$,

$$I_{\text{CPC}} \leq \log N. \qquad (43)$$

*Proof.* We have:

$$I_{\text{CPC}} := \mathbb{E}\left[\frac{1}{N}\sum_{i=1}^{N}\log\frac{f_\theta(x_i,y_i)}{\frac{1}{N}\sum_{j=1}^{N}f_\theta(x_i,y_j)}\right] \tag{44}$$

$$\leq \mathbb{E}\left[\frac{1}{N}\sum_{i=1}^{N}\log\frac{f_\theta(x_i,y_i)}{\frac{1}{N}f_\theta(x_i,y_i)}\right] \leq \mathbb{E}\left[\frac{1}{N}\sum_{i=1}^{N}\log N\right] = \log N \tag{45}$$

which completes the proof. □

For NWJ, we note that the $I_{\text{NWJ}}$ involves a term denoted as $\mathbb{E}_{x,y\sim p(x)p(y)}\left[e^{f_\theta(x,y)}\right]/e$, which could be dominated by rare data-points that have high $f_\theta$ values. Intuitively, this would make it a poor mutual information estimator by optimizing $\theta$. The NWJ estimator may suffer from high variance when the estimator is optimal (Song & Ermon, 2019), this is also empirically observed in Poole et al. (2019). We provide a proof for completeness as follows.

**Proposition 4.** *Assume that $f_\theta$ achieves the optimum value for $I_{\text{NWJ}}$. Then the variance of the empirical NWJ estimator satisfies* $\text{Var}\left(\hat{I}_{\text{NWJ}}\right) \geq \frac{e^{I(X;Y)}-1}{N}$, *where*

$$\hat{I}_{\text{NWJ}} = \frac{1}{N}\sum_{i=1}^{N}[f_\theta(x_i,y_i)] - \frac{e^{-1}}{N}\sum_{i=1}^{N}\left[e^{f_\theta(\bar{x}_i,\bar{y}_i)}\right]$$

*is the empirical NWJ estimator with $N$ i.i.d. samples $\{(x_i,y_i)\}_{i=1}^{N}$ from $p(x,y)$ and $N$ i.i.d. samples $\{(\bar{x}_i,\bar{y}_i)\}_{i=1}^{N}$ from $p(x)p(y)$.*

*Proof.* Let us denote $z_i = \frac{p(x_i,y_i)}{p(x_i)p(y_i)}$. Clearly $\mathbb{E}_{p(x)p(y)}[z_i] = 1$. Then we have:

$$\begin{aligned}
\text{Var}(z_i) &= \mathbb{E}_{p(x)p(y)}\left[z_i^2\right] - \left(\mathbb{E}_{p(x)p(y)}[z_i]\right)^2 \\
&= \mathbb{E}_{p(x)p(y)}\left[z_i^2\right] - 1 \\
&= \mathbb{E}_{p(x)p(y)}\left[\left(\frac{p(x_i,y_i)}{p(x_i)p(y_i)}\right)^2\right] - 1 \\
&= \mathbb{E}_{p(x,y)}\left[\frac{p(x_i,y_i)}{p(x_i)p(y_i)}\right] - 1 \tag{46} \\
&\geq e^{\mathbb{E}_{p(x,y)}\left[\log\frac{p(x_i,y_i)}{p(x_i)p(y_i)}\right]} - 1 = e^{I(X;Y)} - 1 \tag{47}
\end{aligned}$$

where we use Jensen's inequality for $\log$ at the last step.

From Nguyen et al. (2010), we have:

$$f_\theta(x,y) = 1 + \log\frac{p(x,y)}{p(x)p(y)}. \tag{48}$$

for all $x,y$. Since $\{(x_i,y_i)\}_{i=1}^{N}$ (resp. $\{(\bar{x}_i,\bar{y}_i)\}_{i=1}^{N}$) are $N$ datapoints independently sampled from the distribution $p(x,y)$ (resp. $p(x)p(y)$), we have

$$\begin{aligned}
\text{Var}\left(\hat{I}_{NWJ}\right) &= \text{Var}\left(\frac{1}{N}\sum_{i=1}^{N}[f_\theta(x_i,y_i)] - \frac{e^{-1}}{N}\sum_{i=1}^{N}\left[e^{f_\theta(\bar{x}_i,\bar{y}_i)}\right]\right) \\
&\geq \text{Var}\left(\frac{e^{-1}}{N}\sum_{i=1}^{N}\left[e^{f_\theta(\bar{x}_i,\bar{y}_i)}\right]\right) \\
&= \text{Var}\left(\frac{1}{N}\sum_{i=1}^{N}z_i\right) \geq \frac{e^{I(X;Y)}-1}{N} \tag{49}
\end{aligned}$$

which completes the proof. □

---

**Algorithm 1** Construct Chow-Liu Trees with $\mathcal{V}$-Information

---

**Require:** $\mathcal{D} = \{\hat{X}_i\}_{i=1}^m$, with each $\hat{X}_i$ being a set of datapoints sampled from the underlying distribution of random variable $X_i$. The set of function families $\{\mathcal{V}_{i,j}\}_{i,j=1,i\neq j}^m$ between all the nodes.

1: **for** $i = 1, \ldots, m$ **do**
2:     **for** $j = 1, \ldots, m$ **do**
3:         **if** $i \neq j$ **then**
4:             Calculate the edge weight: $e_{i \to j} = \hat{I}_{\mathcal{V}_{i,j}}(X_i \to X_j; \{\hat{X}_i, \hat{X}_j\})$.
5:         **end if**
6:     **end for**
7: **end for**
8: Construct the fully connected graph $G = (V, E)$, with node set $V = (X_1, \ldots, X_m)$ and edge set $E = \{e_{i \to j}\}_{i,j=1,i\neq j}^m$.
9: Construct the maximal directed spanning tree $g$ on $G$ by Chow-Liu algorithm, where mutual information is replaced by $\mathcal{V}$-information.
10: **return** $g$

---

## C  THE NEW ALGORITHM FOR CHU-LIU TREE CONSTRUCTION

See Algorithm 1; $\hat{I}_{\mathcal{V}_{i,j}}(X_i \to X_j; \{\hat{X}_i, \hat{X}_j\})$ denotes the empirical $\mathcal{V}$-information.

## D  DETAILED EXPERIMENTS SETUP

### D.1  CHU-LIU TREE CONSTRUCTION

Figure 2 shows the Chu-Liu tree construction of Simulation-1~Simulation-6. The Simulation-A and Simulation-B in the main body correspond to Simulation-1 and Simulation-4.

**Simulation-1 $\sim$ Simulation-3** :

The ground-truth Chu-Liu tree is a star tree (i.e. all random variables are conditionally independent given $X_1$). We conduct all experiments for 10 times, each time with random simulated orthogonal matrices $\{W_i\}_{i=2}^{20}$. Simulation-1: $X_1 \sim \mathcal{U}(0, 10)$ and $X_i \mid X_1 \sim \mathcal{N}(W_i X_1, 6I), (2 \leq i \leq 20)$; Simulation-2: $X_1 \sim \mathcal{U}(0, 10)$ and $X_i \mid X_1 \sim W_i \mathcal{E}(X_1 + \epsilon_i), (2 \leq i \leq 20), \epsilon_i \sim \mathcal{E}(0.1)$; Simulation-3 is a mixed version:$X_1 \sim \mathcal{U}(0, 10), X_i \mid X_1 \sim \frac{1}{2}\mathcal{N}(W_i X_1, 6I) + \frac{1}{2}W_i \mathcal{E}(X_1 + \epsilon_1), (2 \leq i \leq 20)$.

**Simulation-4 $\sim$ Simulation-6** :

The ground-truth Chu-Liu tree is a tree of depth two. We conduct all experiments for 10 times, each time with random simulated orthogonal matrices $\{W_i\}_{i=2}^7$. Simulation-4: $X_1 \sim \mathcal{U}(0, 10), X_i \mid X_1 \sim \mathcal{N}(W_i X_1, 2I)(i = 2, 3), X_i \mid X_2 \sim \mathcal{N}(W_i X_2, 2I)(i = 4, 5), X_i \mid X_3 \sim \mathcal{N}(W_i X_3, 2I)(i = 6, 7)$; Simulation-5: $X_1 \sim \mathcal{U}(0, 10), X_i \mid X_1 \sim \mathcal{E}(X_1 + \epsilon_i)(i = 2, 3), X_i \mid X_2 \sim W_i \mathcal{E}(X_2 + \epsilon_i)(i = 4, 5), X_i \mid X_3 \sim W_i \mathcal{E}(X_3 + \epsilon_i)(i = 6, 7), \epsilon_i \sim \mathcal{E}(0.1)$; Simulation-6 is a mixed version: $X_1 \sim \mathcal{U}(0, 10), X_i \mid X_1 \sim W_i \mathcal{E}(X_1 + \epsilon_i)(i = 2, 3), X_i \mid X_2 \sim \mathcal{N}(W_i X_2, 2I)(i = 4, 5), X_i \mid X_3 \sim \mathcal{N}(W_i X_3, 2I)(i = 6, 7), \epsilon_i \sim \mathcal{E}(0.1)$.

### D.2  FAIRNESS

We can adapt the $\mathcal{V}$-information perspective to fairness. Denote the random variable that represents sensitive data and the representation as $U$ and $Z$ respectively. Assume $U$ is discrete and $\mathcal{V}$ belongs to preditive family 1. Then we have $H_\mathcal{V}(U) = H(U)$ as long as $\mathcal{V}$ has $\mathrm{softmax}$ on the top and belongs to predictive family. In this case, minimizing $I_\mathcal{V}(Z \to U)$ equals to minimize $-H_\mathcal{V}(Y|X)$. Let the joint distribution of $Z$ and $U$ be paramterized by $\phi$. Hence the final objective is:

$$\min_\phi \{I_\mathcal{V}(u; z)\} = \min_\phi \left( \sup_{f \in \mathcal{V}} \mathrm{E}_{z, u \sim q_\phi(z, u)}[\log P_f(z|u)] \right)$$

In Edwards & Storkey (2015); Madras et al. (2018); Louizos et al. (2015); Song et al. (2018), functions in $\mathcal{V}$ are parameterized by a discriminator.

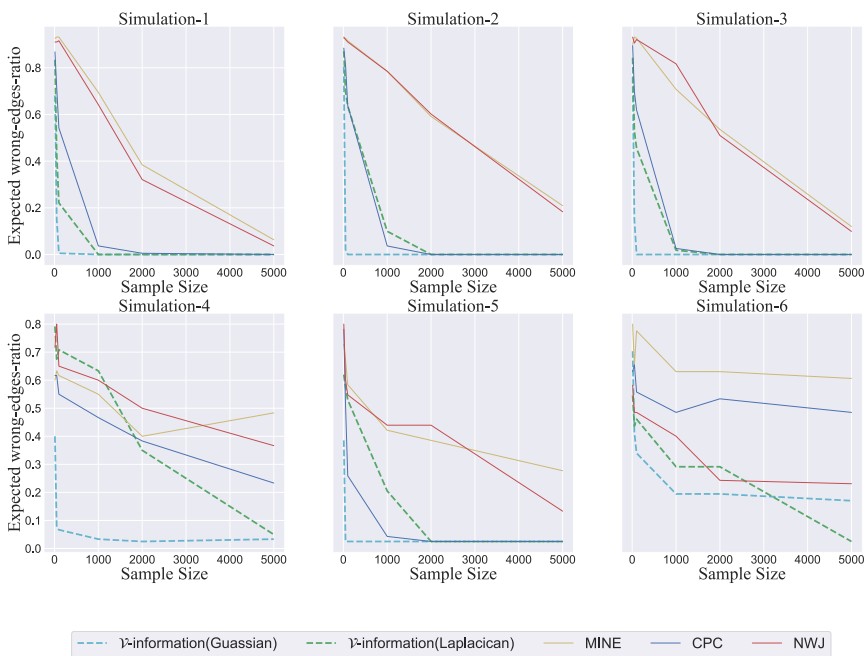

Figure 2: Chu-Liu Tree Construction: The expected wrong-edges-ratio of algorithm 1 with different $\mathcal{V}$ and other mutual information estimators-based algorithms from sample size 10 to $5 \times 10^3$.

For the $(F_i, F_j)$ elements described in the main body, please refer to figure 3b. The three datasets are: the UCI Adult dataset[2] which has gender as the sensitive attribute; the UCI German credit dataset[3] which has age as the sensitive attribute and the Heritage Health dataset[4] which has the 18 configurations of ages and gender as the sensitive attribute.

The models in the figure are:

$\mathcal{V}_A = \{f : \mathcal{Z} \to \mathcal{P}(\mathcal{U}) | f[z](u) = \sum_{(z_i, u_i) \in \mathcal{D}} \frac{e^{\|z_i - z\|_2^2 / h}}{\sum_{(z_i, u_i) \in \mathcal{D}} e^{\|z_i - z\|_2^2 / h}} * \mathbb{I}(u_i = u), h \in \mathbb{R}\}$, where $\mathcal{D}$ is the training set.

$\mathcal{V}_B = \{f : f[z] = \mathrm{softmax}(g(z))\}$, where $g$ is a two-layer MLP with Relu as the activation function.

$\mathcal{V}_C = \{f : f[z] = \mathrm{softmax}(g(z))\}$, where $g$ is a three-layer MLP with LeakyRelu as the activation function.

We further visualize a special case of the $(\mathcal{V}_A, \mathcal{V}_B)$ pair in figure 3a, where the $\mathcal{V}_i = \{f : \mathcal{Z} \to \mathcal{P}(\mathcal{U}) | f[z](u) = \sum_{(z_i, u_i) \in \mathcal{D}} \frac{e^{\|z_i - z\|_2^2 / h}}{\sum_{(z_i, u_i) \in \mathcal{D}} e^{\|z_i - z\|_2^2 / h}} * \mathbb{I}(u_i = u), h \in \mathbb{R}\}$ explicitly makes the features of different sensitivity attributes more evenly spread, and functions in $\mathcal{V}_B$ is a simple two layers MLP with softmax at the top. The leaned features by $\mathcal{V}_A$-information minimization appear more evenly spread as expected, however, the attacker functions in $\mathcal{V}_B$ can still achieve a high AUC of $0.857$.

The $(i, j)$ elements of tables in Figure 3b stand for using function family $\mathcal{V}_i$ to attack features trained with $\mathcal{V}_j$-information minimization. The diagonal elements in the matrix are usually the smallest in rows, indicating that the attacker function family $\mathcal{V}_i$ extracts more information on featured trained with $\mathcal{V}_{j(j \neq i)}$-information minimization.

---

[2]https://archive.ics.uci.edu/ml/datasets/adult
[3]https://archive.ics.uci.edu/ml/datasets
[4]https://www.kaggle.com/c/hhp

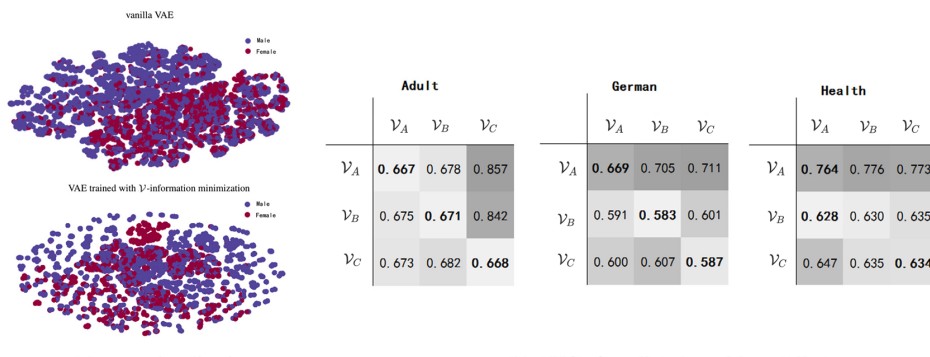

| | **Adult** | | | | **German** | | | | **Health** | | |
|---|---|---|---|---|---|---|---|---|---|---|---|
| | $\mathcal{V}_A$ | $\mathcal{V}_B$ | $\mathcal{V}_C$ | | $\mathcal{V}_A$ | $\mathcal{V}_B$ | $\mathcal{V}_C$ | | $\mathcal{V}_A$ | $\mathcal{V}_B$ | $\mathcal{V}_C$ |
| $\mathcal{V}_A$ | **0.667** | 0.678 | **0.857** | $\mathcal{V}_A$ | **0.669** | 0.705 | 0.711 | $\mathcal{V}_A$ | **0.764** | 0.776 | 0.773 |
| $\mathcal{V}_B$ | 0.675 | **0.671** | 0.842 | $\mathcal{V}_B$ | 0.591 | **0.583** | 0.601 | $\mathcal{V}_B$ | **0.628** | 0.630 | 0.635 |
| $\mathcal{V}_C$ | 0.673 | 0.682 | **0.668** | $\mathcal{V}_C$ | 0.600 | 0.607 | **0.587** | $\mathcal{V}_C$ | **0.647** | 0.635 | **0.634** |

(a) t-sne visualization          (b) AUC of predicted sensitive attribute

Figure 3: T-SNE Visualization and AUC of predicted sensitive attribute

# E  MINIMALITY OF PREDICTIVE FAMILY

Define $\mathcal{V}_{\mathcal{X} \to \mathcal{P}(\mathcal{Y})} = \{g : \mathcal{X} \to \mathcal{P}(\mathcal{Y}) | \exists f \in \mathcal{V}, \forall x \in \mathcal{X}, g[x] = f[x]\}$. Similarly define $\mathcal{V}_{\varnothing \to \mathcal{P}(\mathcal{Y})} = \{g : \varnothing \to \mathcal{P}(\mathcal{Y}) | \exists f \in \mathcal{V}, g[\varnothing] = f[\varnothing]\}$. Intuitively, $\mathcal{V}_{\mathcal{X} \to \mathcal{P}(\mathcal{Y})}$ (resp. $\mathcal{V}_{\varnothing \to \mathcal{P}(\mathcal{Y})}$) restricts the domain of functions in $\mathcal{V}$ to $\mathcal{X}$ (resp. $\varnothing$).

**Non-Negativity**  As we demonstrated in Proposition 2, optional-ignorance guarantees that information will be non-negative for any $X$ and $Y$. Conversely, given any discrete $X, Z, \mathcal{V}_{\varnothing \to \mathcal{P}(\mathcal{Y})}, \mathcal{V}_{\mathcal{X} \to \mathcal{P}(\mathcal{Y})}$ that does not satisfy optional-ignorance, there exists distribution $X, Y$ such that $I_\mathcal{V}(X \to Y) < 0$. Choose $Y \sim f^*[\varnothing]$ where $f^*$ is the function that has no corresponding $g \in \mathcal{V}_{\mathcal{X} \to \mathcal{P}(\mathcal{Y})}$ that can ignore its inputs. Pick $X$ as the uniform distribution, and note that for all $g \in G$, there exists some measurable subset $X' \subset X$ on which $g$ will produce a distribution unequal to $f^*[\varnothing]$, and therefore having higher cross entropy. The expected cross entropy expressed in $H_{\mathcal{V}_{\mathcal{X} \to \mathcal{P}(\mathcal{Y})}}(Y|X)$ is thus higher than in $H_{\mathcal{V}_{\varnothing \to \mathcal{P}(\mathcal{Y})}}(Y)$, and $I_\mathcal{V}(X \to Y) < 0$. Thus, if the function class does not satisfy optional ignorance, then the $\mathcal{V}$-information could be negative.

**Independence**  Given any discrete $X, Y, \mathcal{V}_{\varnothing \to \mathcal{P}(\mathcal{Y})}, \mathcal{V}_{\mathcal{X} \to \mathcal{P}(\mathcal{Y})}$ that does not satisfy optional-ignorance, there exists an independent $X, Y$ such that $I_\mathcal{V}(X \to Y) > 0$. Choose $Y$ such that the distribution $P_Y$ can be expressed as $g[x]$ for some $x \in X, g \in \mathcal{V}_{\mathcal{X} \to \mathcal{P}(\mathcal{Y})}$, but cannot be expressed by any $f \in \mathcal{V}_{\varnothing \to \mathcal{P}(\mathcal{Y})}$. Let $X$ be the distribution with all its mass on $x$; note that the cross entropy of $P_Y$ with $g[x]$ will be zero, and is less than that of the function $f[\varnothing]$ (because $f[\varnothing]$ and $P_Y$ differs on a measurable subset, the cross entropy will be positive). Thus, if the function class does not satisfy optional ignorance, then the $\mathcal{V}$-information does not take value 0 when the two distributions are independent.

# F  LIMITATIONS AND FUTURE WORK

$\mathcal{V}$-information is empirically useful, has several intuitive theoretical properties, but exhibits certain limitations. For example, Shannon information can be manipulated with certain additive algebra (e.g. $H(X, Y) = H(X) + H(Y \mid X)$), while the same does not hold true for general $\mathcal{V}$-Information. However, this could be possible if we choose $\mathcal{V}$ to be a mathematically simple set, such as the set of polynomial time computable functions. It would be interesting to find special classes of $\mathcal{V}$-Information where additional theoretical development is possible.

Another interesting direction is better integration of $\mathcal{V}$-Information with machine learning. The production of usable information (representation learning), acquisition of usable information (active learning) and exploitation of usable information (classification and reinforcement learning) could potentially be framed in a similar $\mathcal{V}$-information-theoretic manner. It is interesting to see whether fruitful theories can arise from these analyses.

