# OpenReview forum: "A Theory of Usable Information under Computational Constraints"
_ICLR.cc/2020/Conference — Accept (Talk)_

### Official Review · AnonReviewer2 · 2019-10-23
**Official Blind Review #2**

**Rating:** 8

**Review:**

Summary
The paper introduces a framework for quantifying information about one random variable, given another random variable (“side information”) and, importantly, a function class of allowed transformations that can be applied to the latter. This matches the typical scenario in machine learning, where observations (playing the role of side information) can be transformed (with a restricted class of transformation-functions) such that they become maximally predictive about another random variable of interest (“labels”). Using this framework, the paper defines the notion of conditional F-entropy and F-entropy (by conditioning on an empty set). Interestingly, both entropic quantities are shown to have many desirable properties known from Shannon entropy - and when allowing the function class of transformations to include all possible models F-entropies are equivalent to Shannon entropies. The paper then further defines “predictive F-information” which quantifies the increase in predictability about one random variable when given side information, under a restricted function-class of allowed transformations of the side information. Importantly, transformations of side information can increase predictive F-information (which is the basis for the notion of “usable” information), which is in contrast to the data processing inequality that applies to Shannon information and states that no transformation of a variable can increase predictability of another variable further than the un-transformed variable (information cannot be generated by transforming random variables). The paper highlights interesting properties of the F-quantities, most notably a PAC bound on F-information estimation from data, which gives reason to expect F-information estimation to be more data-efficient than estimating Shannon-information (particularly in the high-dimensional regime). This finding is confirmed by four types of interesting experiments, some of which make use of a modified version of a tree-structure learning algorithm proposed in the paper (using predictive F-information instead of Shannon mutual information).

Contributions
i) Proposal of a framework for measuring and reasoning about information that transformed random variables have about other random variables, when the class of transformation functions is restricted. Interesting properties are highlighted and corresponding proofs are given. Important conclusions to Shannon-information measures are drawn.

ii) PAC guarantees for estimating F-information quantities from data. A nice result that justifies some optimism about the scalability of F-information estimation.

iii) Modification of a tree-structure learning algorithm, and application to four types of experiments with comparisons against methods for estimating Shannon(-mutual)-information.

Quality, Clarity, Novelty, Impact
The paper is very well written, the motivation and main results are clear and connections to known measures for information in complex systems are drawn (which often appear as corner-cases, or unrestricted cases of F-information). I am not an expert on various information measures, thus I cannot fully judge the novelty of the framework (given that the central idea is fairly simple and quite elegant, the main work lies in the proofs and connections to other frameworks). However, I have not seen the framework being discussed in the machine learning literature before. I personally would rate the potential impact of the F-information framework as high because it addresses many problems that Shannon-(mutual-)information has (hard to estimate, generality means complete blindness against model-classes). The experiments in the paper already illustrate how F-information could be very useful for a range of ML problems that cannot be tackled by strong competitor methods based on Shannon-information estimation. My only criticism is that the paper does not clearly state current limitations and shortcomings and does not comment on the difficulties / potential problems with solving the variational problem that is part of the definition of (conditional) F-information. I currently vote and argue for accepting the paper, though my assessment is of medium confidence only, and I am happy to take issues raised by the other reviewers and the rebuttal into account. I have not checked the proofs in the appendix in great detail.

Improvements
i) Please add a short section of current shortcomings and caveats, especially with regard to applying the methods in practice.

ii) Please comment on solving the variational optimization problem (the infimum) which is part of the definition of (conditional) F-information. In particular, are there any theoretical statements / bounds / etc. to be made for the case where the infimum is not found exactly - does the measure degrade gracefully or can small errors in this optimization lead to wildly varying/divergent F-information? From a practical point-of-view: how was this optimization done in the experiments (particularly when involving a neural network model), how much computational overhead did this optimization add (and how does it compare against other methods, e.g. in terms of wall-clock time or other reasonable metrics, the more the better)?

iii) This is a minor one and feel free to completely ignore it. The name F-information might easily get confused with the use of f-divergences, perhaps there is a better, more informative name. Also, while I personally like the term “usable” in the title, I’m not so sure about “computational constraints” - the latter somehow suggests that the method has small computational footprint, or can easily scale to different computational budget. Perhaps there is a way that more strongly indicates that this refers to restrictions on the model-/function-class (which the term “usable” does already to some degree admittedly).


Minor Comments
a) Have you had any thoughts on how F-information could be used in a rate-distortion / information-bottleneck type framework for a theory of “relevant usable information”? This is probably beyond the scope of this paper, just out of curiosity.

b) The paragraph above 3.3 almost sounds a bit like Shannon (and the data processing inequality) was wrong. I’d rather phrase this as a “no-free-lunch problem” - while the DPI and Shannon (mutual) information is very elegant, it is necessary to make further assumptions/restrictions (the function class of allowed transformations) to make more fine-grained statements and define more precise (but less general) informational-quantities tailored to the specific function class.

c) When choosing function classes that allow for universal function approximation, would F-information degrade to Shannon information?

**Experience Assessment:**

I have read many papers in this area.

**Review Assessment: Checking Correctness Of Derivations And Theory:**

I assessed the sensibility of the derivations and theory.

**Review Assessment: Checking Correctness Of Experiments:**

I assessed the sensibility of the experiments.

**Review Assessment: Thoroughness In Paper Reading:**

I read the paper at least twice and used my best judgement in assessing the paper.

---

> ### Author Response · Authors · 2019-11-10
> **Thank you for your review and suggestions**
>
> Thank you for your review and suggestions.
>
> Q: Please add a short section of current shortcomings and caveats, especially with regard to applying the methods in practice.
>
> Response: We have added a limitations section, reproduced below:
>
> F-information is empirically useful and has appealing theoretical properties. However, some elegant properties of Shannon information are lost. For example, Shannon information can be manipulated with algebra (e.g. H(X, Y) = H(X) + H(Y | X)), while F-Information cannot (for general F). Additionally, for F-Information to be useful in practice, the predictive family F should be easy to optimize over. Machine learning research has identified many such functions (e.g. linear functions, convex functions, ReLU neural networks) suitable for a variety of data types. Nevertheless one should be cautious when applying F-Information to functions and data types that are not well understood in the machine learning literature. In the finite data regime, overfitting is also an issue to consider, and standard techniques to prevent it (e.g., crossvalidation) should be applied.
>
> An interesting direction for future work is to better integrate F-Information with other areas of machine learning. The production of usable information (representation learning), acquisition of usable information (active learning) and exploitation of usable information (classification and reinforcement learning) could potentially benefit from the F-information concept.
>
>
> Q: Please comment on solving the variational optimization problem (the infimum) which is part of the definition of (conditional) F-information.
>
> Response: F-information estimation degrades gracefully with sub-optimal optimization. Let I be the true F-information, I’ be its finite-data estimation with perfect optimization (the infimum is achieved), and I’’ be its estimation with imperfect optimization. Theorem 1 upper bounds | I - I’ |, and we can immediately derive an upper bound on | I - I’’ | by triangle inequality | I  - I’’ | \leq | I - I’ | + | I’ - I’’ |
>
> In other words, the estimation error can only increase by | I’ - I’’ |, which is the gap between perfect optimization and imperfect optimization.
>
> In practice, machine learning research has identified many function families that are empirically easy to optimize (including modern deep neural networks) — which we use as our function family F. We used standard optimization algorithms (e.g. SGD for neural networks) and the wall clock time is identical to other estimators.
>
>
> Q: The name F-information might easily get confused with the use of f-divergences, perhaps there is a better, more informative name.
>
> Response: Thank you for this suggestion. We are considering a name change to V-information as in variational information.
>
>
> Q: Have you had any thoughts on how F-information could be used in a rate-distortion / information-bottleneck type framework for a theory of “relevant usable information”?
>
> Response: The application to information-bottleneck should be straight-forward. In fact, our fairness experiment can be thought of as the opposite of an information bottleneck:
>
> Fairness:			   minimize F information between the learned representation and target (sensitive attributes) and maximize F information w.r.t input.
>
> Information bottleneck: maximize F information between the learned representation and target (labels) and minimize F information w.r.t input.
>
>
> Q: The paragraph above 3.3 almost sounds a bit like Shannon (and the data processing inequality) was wrong. I’d rather phrase this as a “no-free-lunch problem”
>
> Response: We have reworded a few sentences to highlight the no-free-lunch perspective.
>
>
> Q: When choosing function classes that allow for universal function approximation, would F-information degrade to Shannon information?
>
> Response: Yes, this is an expected and desirable property as in Proposition 1. Roughly speaking, if F contains every function and every probability measure — there are no computational constraints — then all information is usable, which is exactly what Shannon information measures. The statistical and computational burden however, makes this a poor design choice for many machine learning problems.

---

> > ### Comment · AnonReviewer2 · 2019-11-13
> > **Thanks for the detailed answers**
> >
> > Thank you for addressing all issues raised in a convincing and thorough manner and preparing a revised manuscript!

---

### Official Review · AnonReviewer1 · 2019-10-27
**Official Blind Review #1**

**Rating:** 8

**Review:**

The paper presents a generalization of classical definitions of entropy and mutual information that can capture computational constraints. Intuitively, information theoretic results assume infinite computational resources, so they may not correspond to how we treat "information" in practice. One example is public-key encryption. An adversary that has infinite time will eventually break the code so the decrypted message conveys the same amount of information (in a classical sense) as the plaintext message. In practice, this depends on computational time.

The authors' approach is to first restrict the class of conditional probability distribution p(Y|X) to a restricted family F that satisfies certain conditions. Unfortunately, the main condition in Def 1 that the authors assume is not natural and is only added to ensure that mutual information remains positive. However, putting this aside, the subsequent definitions that general entropy, conditional entropy, and mutual information are well-motivated.

The authors, then, show that many measures of "uncertainty" can be viewed as "entropies" under this generalized definition including the Mean Absolute Deviation and the Coefficient of Determination.

The overall framework can justify practices that we commonly use in machine learning, which would be justifiable using classical information. One important example is Representation Learning, which is a post-processing of data to aid the prediction task. According to classical information theory, this post-processing shouldn't help because it cannot add more information about the label Y than what was original available in X. Under the formulation presented in this paper, postprocessing can help if we keep in mind information about Y in X are hard to extract to begin with.

In terms of practical applications, the main advantage of the new definition is that F-information can be estimated from a finite sample, simply because F is a restricted set. However, this restriction helps compared to using state-of-the-art estimators for Shannon mutual information as shown in the experiments.

Finally, the literature review section is quite excellent.

I find the overall approach to be quite interesting and definitely worth publishing. The only suggestion I have is that the authors include immediately after Definition 1 a concrete example that illustrates it. For example, suppose that Y is a scalar and X is a noisy estimate of Y. Suppose we restrict F to the family of Gaussian distributions. That is, with side information x, f[x](y)  = N(x, s). Without side information, f[empty](y) = N(u, s). The functions f are parameterized by u and s.
Is this a "predictive family"? To make sure I understand it correctly, can you please walk me through the Eq 1 for this particular example?

Some minor remarks:
- Reference Shannon and Weaver was published in 1963, not 1948.
- In Page 5, "maybe not expressive" should be "may not be expressive".




**Experience Assessment:**

I have published one or two papers in this area.

**Review Assessment: Checking Correctness Of Derivations And Theory:**

I assessed the sensibility of the derivations and theory.

**Review Assessment: Checking Correctness Of Experiments:**

I assessed the sensibility of the experiments.

**Review Assessment: Thoroughness In Paper Reading:**

I read the paper at least twice and used my best judgement in assessing the paper.

---

> ### Author Response · Authors · 2019-11-10
> **Thank you for your review and suggestions**
>
> Thank you for your review and suggestions
>
> Q: Suppose that Y is a scalar and X is a noisy estimate of Y. Suppose we restrict F to the family of Gaussian distributions. That is, with side information x, f[x](y)  = N(x, s). Without side information, f[empty](y) = N(u, s). The functions f are parameterized by u and s.
> Is this a "predictive family"? To make sure I understand it correctly, can you please walk me through the Eq 1 for this particular example?
>
> Response: This is not a predictive family because Eq. 1 doesn’t hold. We can break down Eq 1 for this example into two parts
>
> “For every f \in F, P \in range(f)” translates to —> for any Gaussian distribution N(c, s) where c is any real number
>
> “There exists f’ \in F, f’[x] = P, f’[empty] = P” translates to —> there is an f such that f[empty] = N(c, s), f[x] = N(c, s). In this example, such an f cannot always be found because we are “forced” to use x as the mean of the Gaussian (i.e., we cannot ignore it).
>
> This will be an F information with a small modification: f[x] = N(ax+b, s), f[empty]=N(u, s) where a, b, u are parameters we can optimize. To check Eq 1 we can verify
>
> “There exists f’ \in F, f’[x] = P, f’[empty] = P” -> this can be achieved by choosing a=0, b=c, u=c
>
> In fact, this quantity is equal to the R^2 coefficient (Proposition 1.5) — a common measurement of dependence between two random variables.
>
> Note that many nice properties continue to hold, but without Eq. 1 F-information can be negative.
>
>
> Q: Reference Shannon and Weaver was published in 1963, not 1948.  In Page 5, "maybe not expressive" should be "may not be expressive".
>
> Response: Thank you for the correction. We have fixed them.

---

### Decision · Program_Chairs · 2019-12-19

**Decision:**

Accept (Talk)

**Comment:**

All reviewers unanimously accept the paper.